# HelpSteer3-Preference: Open Human-Annotated Preference Data across Diverse Tasks and Languages

**Zhilin Wang, Jiaqi Zeng, Olivier Delalleau, Hoo-Chang Shin,**
**Felipe Soares, Alexander Bukharin, Ellie Evans, Yi Dong, Oleksii Kuchaiev**
NVIDIA
{zhilinw, jiaqiz}@nvidia.com

## Abstract

Preference datasets are essential for training general-domain, instruction-following language models with Reinforcement Learning from Human Feedback (RLHF). Each subsequent data release raises expectations for future data collection, meaning there is a constant need to advance the quality and diversity of openly available preference data. To address this need, we introduce HelpSteer3-Preference, a permissively licensed (CC-BY-4.0), high-quality, human-annotated preference dataset comprising of over 40,000 samples. These samples span diverse real-world applications of large language models (LLMs), including tasks relating to STEM, coding and multilingual scenarios. Using HelpSteer3-Preference, we train Reward Models (RMs) that achieve top performance on RM-Bench (82.4%) and JudgeBench (73.7%). This represents a substantial improvement (∼10% absolute) over the previously best-reported results from existing RMs. We demonstrate HelpSteer3-Preference can also be applied to train Generative RMs and how policy models can be aligned with RLHF using our RMs.

🤗 **Dataset (CC-BY-4.0):** huggingface.co/datasets/nvidia/HelpSteer3#preference
🤗 **Models (NV Open Model):** huggingface.co/collections/nvidia/reward-models

## 1 Introduction

| Dataset | Released | Quality | Diverse | Multilingual | Annotator | Commercial Use |
|---|---|---|---|---|---|---|
| HH-RLHF [1] | Apr 22 | ✗ | ✗ | ✗ | 🧑 | ✓ |
| Open Assistant [2] | Apr 23 | ✗ | ✗ | ✓ | 🧑 | ✓ |
| UltraFeedback [3] | Oct 23 | 🟡 | 🟡 | ✗ | 🤖 | 🟡 |
| HelpSteer [4] | Nov 23 | 🟡 | 🟡 | ✗ | 🧑 | ✓ |
| Nectar [5] | Nov 23 | 🟡 | 🟡 | ✗ | 🤖 | 🟡 |
| Skywork-Preference [6] | Oct 24 | ✓ | 🟡 | ✗ | 🧑+🤖 | 🟡 |
| HelpSteer2-Preference [7] | Oct 24 | ✓ | 🟡 | ✗ | 🧑 | ✓ |
| INF-ORM-Preference [8] | Dec 24 | ✓ | 🟡 | ✗ | 🧑+🤖 | 🟡 |
| **HelpSteer3-Preference (Ours)** | Mar 25 | ✓ | ✓ | ✓ | 🧑 | ✓ |

Table 1: Comparison of Helpsteer3-Preference with select General-Domain Preference Datasets. See main text for rationales for the classification into good (✓), moderate (🟡) and poor (✗) on all aspects.

Reinforcement Learning from Human Feedback (RLHF) has been an important part of training general-domain instruction-following language models, starting from early works such as InstructGPT [1] and HH-RLHF [9] to state of the art open-weight models from DeepSeek [10], Llama [11] and Qwen [12]. Despite recent efforts to perform reinforcement learning with rule-based rewards [13, 14] for specific domains containing verifiable ground-truth answers such as math, logical reasoning and

competitive programming, these works also stress the importance for human feedback for other domains where model responses cannot easily be verified for correctness. For instance, when asking a language model to write an engaging story, there remains many aspects of human preference (e.g. creativity, pace or plot) that cannot be easily verified. A critical ingredient to support the research community in applying RLHF and exploring new methods and algorithms is thus the availability of high-quality, diverse and commercial-friendly preference datasets that can be used to train general-domain instruction-following language models.

HH-RLHF [1] and Open Assistant [2] represent the first generation of general-domain preference datasets, distinct from the earlier specific-domain preference data such as OpenAI Summarize [15] (for summarization), Stanford Human Preferences [16] (for online forums) and Stack Exchange Preference [17] (for software engineering). As initial attempts into building general-domain preference datasets, HH-RLHF/Open Assistant suffer from low-quality data [18, 19] as they use model responses from a limited selection of relatively weak models (by May 2025 standards) [1] and/or employ crowd-workers/volunteers with limited quality controls [1, 2]. In addition, they typically use simple prompts with limited diversity [20, 21], reflecting the limited LLM capabilities when such datasets were built. HH-RLHF has samples in English only while Open Assistant is multilingual. Both datasets are permissively licensed (MIT or Apache 2.0), enabling commercial use with minimal restrictions.

To overcome the limitation of low-quality human annotations, the next generation of general-domain preference datasets (UltraFeedback [3], HelpSteer [4] and Nectar [5]) used two distinct strategies. UltraFeedback and Nectar chose to use a strong LLM (GPT-4) as an annotator, replacing human annotators used by HH-RLHF and Open Assistant. While this does improve quality relative to low-quality human annotations, this also means that the quality of the dataset is bottle-necked by GPT-4's accuracy as an LLM-annotator, which is only 86.0% on RewardBench [22] and substantially behind top performing Reward Models trained on high-quality human-annotated data. GPT-4 is also known to have self-enhancement bias [23, 24], which means that it prefers its own responses and those from models trained to respond similarly through methods such as distillation. Furthermore, the use of GPT-4 annotations means that use of such preference datasets can be limited in some commercial settings due to OpenAI's terms of use [25]. On the other hand, HelpSteer [4] continued to use human annotators, but added more quality-control methods to reduce the occurrence of low-quality annotations. These methods include giving more comprehensive guidelines and examples to annotators, providing more training to annotators (giving a training course and preliminary assignments, which only those who pass can join the project), and doing more post-annotation quality assurance (manual reviews in addition to automated checks). The choice of data source and human annotation (instead of GPT-4 annotation) also facilitated HelpSteer's release with a permissive CC-BY-4.0 license [4]. All three datasets also improve diversity by using user-contributed/generated prompts from diverse sources including ShareGPT [26], UltraChat [27], Evol-Instruct [28], FLAN [29], TruthfulQA [30], FalseQA [31], HH-RLHF [1] and LMSysChat-1M [32]. Nonetheless, Ultrafeedback, HelpSteer and Nectar only contain English samples.

The third generation of general-domain preference datasets (Skywork-Preference [6], HelpSteer2-Preference [7], INF-ORM-Preference [8]) seek to further improve quality. Skywork-Preference and INF-ORM-Preference do so by combining select high-quality preference datasets (annotated either by humans or LLMs) including HelpSteer2 [18], Magpie [33], OffsetBias [34] and WildGuard [35]. On the other hand, HelpSteer2-Preference improves quality by provisioning more stringent data annotation practices (3-5 independent annotations per sample) and data filtering approaches (removing samples with large disagreements and filtering outlier annotations). All three datasets also contain English prompts only.

While there has been substantial progress made in terms of data quality, improvements in data diversity have been more limited. Diversity is important because as developments in LLMs make them more capable, they are also used for an increasing variety of challenging tasks [36]. To ensure that RLHF can still be effective on such difficult tasks, preference datasets need to incorporate a diverse spread of challenging tasks. For instance, while LLMs have been previously used to generate short code snippets [37, 38], they have recently been used for more complex coding and debugging scenarios requiring substantially longer responses [39]. Additionally, diverse language representation in publicly-available post-training datasets is a known challenge [40], especially for general-domain preference datasets among which only Open Assistant [2] is multilingual. As LLMs are increasingly adopted by users with different native languages, expanding preference datasets to cover languages beyond English becomes critical to training LLMs well for these users.

To provide effective feedback on such diverse, challenging and multilingual tasks, we need specialist annotators that have deep expertise in various topics and languages. While it might be possible to use LLMs to role-play specialist-annotators [41, 42], we are not aware of any evidence on how well they can provide preference feedback relative to human specialist annotators. Nonetheless, the reality that top proprietary LLM providers such as OpenAI, Anthropic and xAI are still hiring for human specialists (directly [43] or through vendors [44, 45]) to perform data annotation, despite having access to the strongest LLMs, indicates that human-specialist annotated data remains valuable.

We collect HelpSteer3-Preference by engaging specialist pools of annotators to perform high-quality annotation on various task categories. These categories include STEM (Science, Technology, Engineering and Mathematics), Code and Multilingual, in addition to the General tasks, as covered in HelpSteer2-Preference previously. These specialist annotator pools require a higher bar for inclusion into the project, such as requiring a degree in relevant fields, work experience and proficiency in specific languages. These annotators indicate preference on diverse real-world tasks from WildChat-1M [46] and ShareGPT [26] with responses generated by 17 language models, resulting in over 40 thousand samples for preference modeling. Using HelpSteer3-Preference, we train SOTA reward models on RM-Bench, a popular benchmark for measuring the capabilities of reward models as well as JudgeBench, a benchmark for evaluating LLM-as-a-Judge applications. Our best Reward Models trained on top of Llama-3.3-70B-Instruct reach 82.4% on RM-Bench and 73.7% on JudgeBench, which represent substantial improvements (∼10% absolute) over the current top-performing reward models. We demonstrate that performance can be further boosted by training Generative Reward Models using the same data. Finally, we show how our best reward models can be used for RLHF to align instruction-following models. We openly release HelpSteer3-Preference with CC-BY-4.0 license at huggingface.co/datasets/nvidia/HelpSteer3#preference.

## 2  Dataset

### 2.1  Dataset Construction

We use prompts and responses from the HelpSteer3 Feedback dataset [47, 48] (CC-BY-4.0 license). We highlight below relevant aspects of the prompt/response sourcing approach and direct interested readers to [47] for details. Our main contribution lies in the collection and open release of preference annotations for these samples.

**Prompt Collection**    Prompts for the Code and Multilingual subsets were taken from the ShareGPT [26] dataset (CC0-1.0 license), as inspired by HelpSteer2-Preference [7], which previously excluded code and multilingual prompts. To avoid potential overlap with prompts from HelpSteer2-Preference dataset, prompts for General and STEM were curated from WildChat-1M [46, 49] (ODC-BY license) instead. WildChat-1M is similar to ShareGPT, except that it is much larger (1 million versus 90 thousand) and were collected more recently (ending in April 2024 as opposed to April 2023). We stratified sampled Code and Multilingual prompts from ShareGPT based on language, while prompts from WildChat were chosen with stratified sampling based on topic and prompt complexity.

**Response Generation**    Responses were generated from a variety of 17 commercially-permissive models that were popular at the start of our annotation (Aug 2024). The chosen models are: Nemotron 4 (340B Instruct) [50], Gemma (2B) [51], Gemma 2 (2B, 9B and 27B) [52], Mistral (7B-Instruct-v0.3 [53], Mistral-Nemo 12B [54], Codestral 22B [55], Mixtral 8x7B Instruct [56], Mixtral 8x22B Instruct [57], Mistral Large 2[58])[1], Phi 3 (Mini, Small and Medium) [59] , IBM Granite (8B and 34B) [60], and Snowflake Arctic [61] families. Responses to the same prompt were generated using two different models. Tasks with potentially unsafe prompts/responses (profanity, harmful and illegal content, responses with bias and stereotypes as well as Personally Identifiable Information) were filtered out using a combination of automated and manual approaches.

**Multi-turn Prompt In-filling**    To include preference pairs on follow-up assistant turns beyond the initial turn, we included multi-turn conversations as context for preference annotations, similar to HelpSteer2-Preference [7]. Seeking to avoid using any ChatGPT-generated assistant turns (even

---

[1]Codestral 22B and Mistral Large 2 were used following a contractual agreement with Mistral to use and release the responses by these models under a commercially permissive license, which we chose as CC-BY-4.0.

within the context), we used the models above (for response generation) to generate intermediate assistant turns. Contexts are restricted to a maximum of 2000 words. Following HelpSteer2-Preference [7], we asked for preference annotations only for the last assistant turn (with annotators only seeing the newly generated intermediate turns, and not the original ones).

**Preference Annotation**   Following HelpSteer2-Preference [7], we required 3-5 independent annotators per sample. For each sample, we asked annotators to choose among the following options, alongside a brief justification of their choice within 1 to 2 sentences (i.e. in 10-50 English words).

   -3.  Response 1 is much better than Response 2 ($A >>> B$)

   -2.  Response 1 is better than Response 2 ($A >> B$)

   -1.  Response 1 is slightly better than Response 2 ($A > B$)

   1.  Response 2 is slightly better than Response 1 ($A < B$)

   2.  Response 2 is better than Response 1 ($A << B$)

   3.  Response 2 is much better than Response 1 ($A <<< B$)

   -100.  Neither response is valid

Annotators for the General, STEM and Code subsets were sourced and managed by Scale AI while annotators for the Multilingual subset were sourced and managed by Translated. Subsets other than General have higher bars for annotator inclusion (i.e. Degree in relevant subjects for STEM, Software Engineering work experience for Code and Language Fluency for Multilingual). In total, over 6400 annotators from 77 countries/regions participated (details in Appendix D). We built upon the annotator guidelines from HelpSteer2-Preference [7] with our guidelines available in Appendix E. For the Multilingual subset, we specifically ask annotators to penalize responses that do not respond with the expected language (typically the prompt language unless otherwise specified) while for Code, we ask annotators to assess whether responses have sufficient comments within the code snippet (for readability) as well as whether the code follows established coding styles [62], beyond the standard guidelines. Annotators are encouraged to use internet search to support their assessment of preferences (e.g. fact checking claims) but prohibited from using LLM tools.

**Preference Post-Processing**   Following HelpSteer2-Preference [7], we remove samples with at least one annotation as 'Neither response is valid' and filter out outlier annotations within the same task, to retain the three annotations that agree most with each other (in terms of their preference strength). Samples that exhibit a large disagreement (>2) among the three most agreeing annotations were excluded, as this suggests that the task was overly subjective or that annotators missed important aspects of consideration in assessing preference. Overall preference is calculated using the average of the three most agreeing preference scores and then rounding to the nearest integer. Therefore, a portion of the samples will have an overall preference of 0: $A = B$ (as some annotations are +1: $A < B$ and others -1: $A > B$). We include such samples in our data release as well as the analysis below, but do not use them during reward model training.

## 2.2   Dataset Analysis

| | N | Context Turns | Context Chars | Response Chars | Weighted Cohen's $\kappa$ | Mean Preference |
|---|---|---|---|---|---|---|
| HelpSteer3-Preference | 40476 | 3.5 (4.1) | 2638 (3579) | 1695 (1253) | 0.890 | -0.003 (1.950) |
| - General | 18638 | 3.2 (3.7) | 2545 (3461) | 1734 (1288) | 0.896 | 0.030 (2.042) |
| - STEM | 4918 | 3.6 (4.2) | 2588 (3520) | 1700 (1152) | 0.894 | 0.023 (2.056) |
| - Code | 8857 | 3.7 (3.8) | 3343 (3973) | 2101 (1218) | 0.897 | -0.128 (1.923) |
| - Multilingual | 8063 | 4.0 (5.1) | 2110 (3295) | 1157 (1062) | 0.857 | 0.042 (1.700) |
| HelpSteer2-Preference | 9125 | 2.8 (3.7) | 711 (876) | 1483 (1063) | 0.878 | 0.065 (1.719) |

Table 2: Descriptive Statistics for HelpSteer3-Preference compared with HelpSteer2-Preference. Numbers written as "X (Y)" refer respectively to mean X and standard deviation Y.

**Improvements over HelpSteer2-Preference**   As shown in Table 2, HelpSteer3-Preference has a substantial proportion of specialist-annotated samples from the STEM (12.2%), Code (21.9%) and Multilingual (19.9%) subsets alongside the General subset (46.0%). Summing across various subsets,

HelpSteer3-Preference (40476 samples) is more than four times the size of HelpSteer2-Preference (9125 samples). Furthermore, HelpSteer3-Preference has context conversations with more turns (3.5 vs. 2.8) and characters (2638 vs. 711) compared to HelpSteer2-Preference. Responses also tend to be slightly longer (1695 vs 1483 characters). The code subset has the longest average length (2101 characters), as responses typically contain both code blocks and textual explanations. The Multilingual subset has the shortest responses (1157 characters), which can be attributed to the non-Latin scripts used for some languages such as Chinese and Korean.

**Diverse Programming and Natural Languages**   Table 3 shows that the Code and Multilingual subsets of HelpSteer3-Preference contain 14 programming and 13 natural languages, respectively. These languages are commonly used languages with a proportion that is representative of their presence in the ShareGPT [26] prompt dataset. Code subset is dominated by Python (38.2%) followed by JS/HTML/CSS (combined as they commonly co-occur) and a long tail of other popular languages. Similarly, the Multilingual subset is most represented by Chinese (30.2 %), followed by Korean, French, Spanish and other widely-used languages.

| *Subset* | | | | | | % of Samples by Language | | | | | | | | |
|---|---|---|---|---|---|---|---|---|---|---|---|---|---|---|
| **Code** | Python | JS/HTML/CSS | C# | SQL | Java | C++ | Go | C | PHP | TS | PowerShell | Rust | R | Bash |
| Proportion (%) | 38.2 | 23.3 | 5.5 | 5.2 | 5.1 | 5.0 | 3.6 | 3.3 | 3.3 | 3.1 | 1.2 | 1.1 | 1.1 | 1.0 |
| **Multilingual** | Chinese | Korean | French | Spanish | Japanese | German | Russian | Port. | Ital. | Viet. | Dutch | Polish | Indonesian | |
| Proportion (%) | 30.2 | 10.4 | 10.1 | 10.1 | 7.0 | 6.2 | 6.0 | 5.6 | 5.2 | 2.4 | 2.4 | 2.2 | 2.1 | |

Table 3: Languages in Code and Multilingual subsets. Chinese tasks include samples in both Simplified Chinese and Traditional Chinese, which we did not distinguish at prompt selection.

**High Inter-Rater Reliability**   HelpSteer3-Preference subsets also consistently show high inter-rater reliability measured with quadratic-weighted Cohen's $\kappa$ [63] as shown in Table 2. Following [7], we used the weighted variant of Cohen's $\kappa$ in order to penalize larger disagreements (e.g. -3: $A >>> B$ and +3: $A <<< B$ ) much more heavily compared to smaller disagreements (e.g. -1 $A > B$ and +1: $A < B$). Weighted Cohen's $\kappa$ is greater than 0.8 for each subset, suggesting strong inter-rater reliability [63]. We account the high reliability to stringent annotator recruitment criteria (to filter out under-qualified annotators), quality control (to filter out under-performing annotators) and post-processing (to filter out outlier annotations and high-disagreement samples).

**Low Position Bias**   Finally, the mean preference within each subset is also low (relative to the standard deviation), indicating a low position bias. Such a slight position bias can be caused by uneven sampling of models at different positions or possibly by annotator biases. Nonetheless, such biases are comparable to those in HelpSteer2-Preference and small compared to LLM-as-a-judge preferences [23].

**Preference Distributions**   To better understand the distribution of preferences, Fig. 2 shows the distribution of each preference label across different HelpSteer3-Preference subsets alongside HelpSteer2-Preference for comparison. The General, STEM and Code subsets show a similar bimodal distribution with peaks near both -2: $A >> B$ and +2: $A << B$. On the other hand, the Multilingual subset shares a similar unimodal distribution as HelpSteer2-Preference with a peak near 0: $A = B$. We hypothesize that this might be because General, STEM and Code subsets were annotated by the same vendor (Scale AI), which might explicitly or implicitly train annotators to give more strong judgments (Response 1 is better/much better than Response 2 or vice-versa) while Multilingual was done by a separate vendor (Translated) whose annotators more frequently give hedged judgments (Response 1 is slightly better than Response 2, or vice-versa). It is also possibly due to the difference in prompt difficulty: Multilingual prompts might be easier as they were sourced from ShareGPT similar to HelpSteer2-Preference. Specifically, ShareGPT consists of user-volunteered prompts to ChatGPT before Apr 2023 when models were generally weaker (especially in multilingual settings), and users might have managed their expectations by sending relatively simpler multilingual prompts. As a result, there could be smaller differences in performance between different models on such simpler prompts. An analysis of the corresponding preference justifications can be found in Appendix G.

# 3 Reward Models

## 3.1 Evaluation

**RewardBench is no longer relevant for evaluating recent top-performing reward models** despite having been a popular benchmark for reward models [64]. The first reason is that RewardBench contains several artifacts that might bias the assessment of reward models. Such artifacts include the chosen response to math prompts having answers in \boxed{} while the rejected responses having answers after # Answer [65]. Another example of such an artifact is the use of GPT-4 to determine ground-truth chosen/rejected responses to some Chat-Hard prompts, biasing RewardBench to reward models trained on GPT-4 generated preference data [7]. Such artifacts make it difficult for RewardBench to serve as a fair evaluation of Reward Models. The second reason is that RewardBench is becoming saturated with top performing models exceeding 95% in accuracy [22]. This means that there remains very little room for the strongest reward models to improve on this benchmark, without potentially overfitting to it. Similar to benchmarks such as MMLU [66] in assessing general knowledge and GSM8K [67] in assessing mathematical ability of LLMs, such performance saturation signals the need for more effective reward model evaluation.

**RM-Bench** [68] is a promising drop-in replacement for RewardBench as it contains similar categories as RewardBench (Chat, Safety, Math and Code) while addressing some of its issues discussed above. Specifically, it increases in difficulty, meaning that the top performing model only reaches 70.1% overall accuracy and only 56.1% accuracy for the Hard subset. Such a dataset presents renewed challenge for training stronger reward models. RM-Bench is also designed to avoid biases suffered by RewardBench. For instance, rather than constructing the chat subset using responses from different models (which can create style biases, as discussed above), it only samples from a single strong model (GPT-4o) to generate chosen responses before injecting targeted errors (with as few as one single word of difference) to create rejected responses. These responses are then verified by humans to be correct and incorrect respectively in order to construct a robust benchmark.

**JudgeBench** [69] is a popular benchmark for measuring models in their ability to act as judges to differentiate correct and incorrect responses relating to General Knowledge, Logical Reasoning, Math and Coding. While JudgeBench can be used to assess different types of models, including prompted LLMs and finetuned LLMs, Reward Models constitute an important class of models to measure such capabilities. This is because Reward Models can be substantially (> 100x) more compute-efficient compared to LLMs of similar size as Reward models only require the equivalent of 1 generated token while LLMs may generate hundreds of tokens in order to make a judgment. JudgeBench is also a challenging benchmark, with the top performing Reward Model only reaching 64.3% accuracy.

## 3.2 Training

**Bradley-Terry/Conventional Reward Models** were trained using the Scaled Bradley-Terry Loss, which has been shown by [7] to work better than regular Bradley-Terry Loss [1, 9] and Bradley-Terry Loss with margin term [70]. Specifically, we train Reward Models initialized from Llama-3.3-70B-Instruct [71] (an updated version of Llama-3.1-70B-Instruct used by [7]) and a feedforward layer that converts the hidden representation of the end-of-response token to a scalar reward. We also train reward models with strong baseline datasets including HelpSteer2-Preference [7], Skywork-Preference (v0.2) [6] and INF-ORM-Preference [8]. Further details are provided in Appendix H.

**Generative Reward Models** have recently emerged as an alternative paradigm to Bradley-Terry models. These models first generate textual critiques of a response and then produce a score based on such critique [72–76]. We adopt a similar reinforcement learning approach as DeepSeek-GRM [76], as it was shown to perform better than other generative methods. Technical details can be found in Appendix H. After unsuccessfully trying with Llama-3.3-70B-Instruct as an initial model, we identify that the generative RM approach requires models to think/reason before responding, hence we use a related reasoning model: Llama-3.3-Nemotron-Super-49B-v1 [77] (see Appendix I).

## 3.3 Results

**Overall** Among Bradley-Terry models, two models are the most promising. The first model is trained on the multilingual subset only, reaching the highest on RM-Bench (82.4%), shown in Table 4.

| | **RM-Bench** | | | | | | | | **JudgeBench** | | | | |
| *Model* | Chat | Math | Code | Safety | Easy | Normal | Hard | **Overall** | Knowl. | Reason. | Math | Coding | **Overall** |
|---|---|---|---|---|---|---|---|---|---|---|---|---|---|
| ***Bradley-Terry Reward Models*** | | | | | | | | | | | | | |
| English RM (General + STEM + Code) | 75.4 | **84.5** | **69.3** | 90.4 | 92.1 | **85.7** | 71.1 | 79.9 | 70.8 | **76.5** | 82.1 | 66.7 | **73.7** |
| Multilingual RM | **86.2** | 82.4 | 66.8 | 94.1 | 86.5 | 85.4 | **80.0** | 82.4 | 66.2 | 71.4 | 82.1 | 59.5 | 69.4 |
| ***Data Ablations*** | | | | | | | | | | | | | |
| All HelpSteer3-Preference subsets | 73.6 | 82.7 | 66.1 | 91.4 | 89.4 | 84.3 | 71.9 | 78.5 | 63.0 | 69.4 | 82.1 | **71.4** | 68.9 |
| General + Code + Multilingual | 72.5 | 75.8 | 66.6 | 93.4 | 90.5 | 82.7 | 66.1 | 77.1 | 68.2 | 72.5 | 78.6 | 61.9 | 70.3 |
| General + STEM + Multilingual | 67.7 | 82.4 | 68.4 | 88.0 | 88.0 | 83.9 | 69.3 | 76.6 | 65.6 | 71.4 | **83.9** | **71.4** | 70.9 |
| STEM + Code + Multilingual | 74.0 | 74.1 | 66.2 | 94.5 | 91.2 | 82.5 | 64.9 | 77.2 | 68.8 | 71.4 | 78.6 | 61.9 | 70.3 |
| General only | 71.9 | 79.8 | 63.1 | 93.5 | 90.4 | 84.3 | 67.5 | 77.1 | 63.6 | **76.5** | 82.1 | 59.5 | 69.7 |
| STEM only | 70.9 | 79.9 | 68.5 | 94.3 | 90.6 | 84.8 | 70.1 | 78.4 | 63.0 | 75.5 | 75.0 | 66.7 | 68.9 |
| Code only | 67.3 | 71.4 | 65.5 | 89.8 | **94.0** | 82.1 | 52.2 | 73.5 | **74.0** | 69.4 | **83.9** | 54.8 | 72.0 |
| ***External Datasets*** | | | | | | | | | | | | | |
| HelpSteer2-Preference | 77.6 | 74.1 | 63.7 | 93.8 | 91.0 | 82.1 | 64.5 | 77.3 | 66.9 | 69.4 | 82.1 | 54.8 | 68.6 |
| INF-ORM-Preference | 76.8 | 72.1 | 61.7 | 94.8 | 89.6 | 80.8 | 64.5 | 76.3 | 63.6 | 70.4 | 80.4 | 52.4 | 66.9 |
| Skywork-Preference | 70.3 | 69.3 | 60.6 | 94.2 | 86.5 | 78.7 | 63.3 | 73.6 | 64.9 | 64.3 | 71.4 | 57.1 | 64.9 |
| ***External Baselines*** | | | | | | | | | | | | | |
| Llama-3.1-Nemotron-70B-Reward | 70.7 | 64.3 | 57.4 | 90.3 | 92.2 | 76.8 | 48.0 | 70.7 | 62.3 | 72.5 | 76.8 | 57.1 | 66.9 |
| Skywork-Reward-Gemma-2-27B* | 71.8 | 59.2 | 56.6 | 94.3 | 89.6 | 75.4 | 50.0 | 70.5 | 59.7 | 66.3 | **83.9** | 50.0 | 64.3 |
| Skywork-Reward-Llama-3.1-8B* | 69.5 | 60.6 | 54.5 | **95.7** | 89.0 | 74.7 | 46.6 | 70.1 | 59.1 | 64.3 | 76.8 | 50.0 | 62.3 |
| ***Generative Reward Models (non-BT)*** | | | | | | | | | | | | | |
| All HelpSteer3-Preference subsets | 73.7 | 91.4 | 75.0 | 90.6 | 91.2 | 85.7 | 71.2 | 82.7 | **71.4** | 73.5 | 87.5 | 76.2 | 75.1 |
| + *voting@32* | 74.0 | 92.7 | 77.4 | 92.1 | **92.6** | 87.3 | 72.3 | 84.0 | 70.8 | **83.7** | 87.5 | 83.3 | **78.6** |
| English RM (General + STEM + Code) | 73.6 | 92.7 | 74.7 | 90.4 | 90.5 | 85.9 | 72.1 | 82.8 | 67.5 | 71.4 | 83.9 | 85.7 | 73.4 |
| + *voting@32* | 73.6 | **93.5** | 77.8 | 91.4 | 91.5 | 87.4 | 73.2 | 84.1 | 66.9 | 79.6 | **92.9** | **88.1** | 77.1 |
| Multilingual RM | **77.2** | 91.9 | 74.7 | 92.9 | 90.7 | 86.7 | 75.1 | 84.2 | 64.9 | 74.5 | 87.5 | 73.8 | 72.3 |
| + *voting@32* | 76.3 | 93.2 | **79.0** | **93.5** | 92.1 | **88.5** | **75.9** | **85.5** | 65.6 | 82.7 | 87.5 | 85.7 | 76.3 |
| HelpSteer2-Preference | 69.9 | 91.2 | 73.9 | 92.0 | 90.7 | 84.6 | 69.9 | 81.7 | 63.6 | 76.5 | 80.4 | 81.0 | 72.0 |
| + *voting@32* | 71.9 | 92.5 | 76.6 | 92.6 | 91.8 | 86.9 | 71.4 | 83.4 | 66.9 | 78.6 | 85.7 | **88.1** | 75.7 |

Table 4: Performance of Reward Models on RM-Bench and JudgeBench. Higher is better for each category. Skywork-Reward-Llama-3.1-8B* and Skywork-Reward-Gemma-2-27B* are the top reward models on the original RM-Bench leaderboard [78] and JudgeBench leaderboard [79] respectively.

The second model is an English Reward Model trained on the General, STEM and Code subsets, achieving the highest on JudgeBench (73.7%) in Table 4 and the second highest on RM-Bench (79.9%). Each of these models shows close to a **10% increase in accuracy** compared to the top-performing reward models reported on RM-Bench [68] and JudgeBench [69] papers. To put this 10% gain into perspective, the improvement on RM-Bench (70.1 to 82.4%) is larger than the gap between the reported Top-20 reward models (62.7% to 70.1%) [68] while the improvement on JudgeBench (64.3% to 73.7%) is larger than the gap between the best-performing and worst-performing reported Reward Models (59.4% to 64.3%) [69]. Compared to the same initial model trained on baseline datasets (HelpSteer2-Preference, INF-ORM-Preference and Skywork-Preference), our best performing models trained on HelpSteer3-Preference are also substantially better, reaching more than 5% higher accuracy than models trained on the best baseline dataset on both RM-Bench and JudgeBench.

**English vs. Multilingual RM** While the Multilingual RM performs better than English RM on RM-Bench Overall (82.4% vs 79.9%), their performance gap is greatly modulated by the difficulty of the questions. On Easy prompts, English RM performs much better (92.1% vs. 86.5%) while on Hard problems, Multilingual RM performs much better (80.0% vs 71.1%). In RM-Bench, difficulty refers to whether the preferred response has additional stylistic advantages relative to the dis-preferred response. Specifically, if the preferred response has more verbosity (i.e. higher response length) and/or markdown formatting (i.e. markdown elements such as headings, bolded and lists) while the dis-preferred response does not, the problem is considered to be easy. If the dis-preferred response has more verbosity and/or markdown formatting than the preferred response, then the problem is considered to be difficult. Therefore, the difference in performance reflects how the reward models account for stylistic factors such as response length and markdown features.

Fig. 1 shows that the English RM is substantially more influenced by response length than Multilingual RM. For English RM, comparing the concise chosen response ($y_c^{\emptyset}$) with the concise rejected response ($y_r^{\emptyset}$) achieves a win rate of 84.6% which drops drastically by 24.8% to 59.8% when the same concise chosen response ($y_c^{\emptyset}$) is compared with the verbose rejected response ($y_r^{L}$). For Multilingual RM, the drop for the same comparison is much lower at only 6.4% (from 84.7 to 78.3%). The English RM is also slightly more influenced by Markdown formatting than Multilingual RM. For English RM, comparing the verbose chosen response ($y_c^{L}$) with the verbose rejected response ($y_r^{L}$) achieves

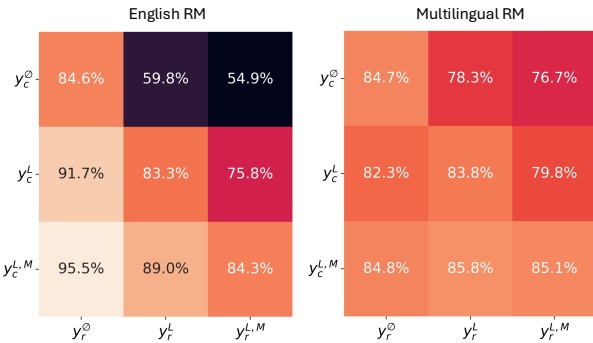

Figure 1: Win rate by setting for English and Multilingual RM. $y_c$ refers to the chosen response while $y_r$ refers to the rejected response. $y^{\emptyset}$ refers to the concise response, $y^L$ refers to the verbose response while $y^{L,M}$ refers to the verbose response with markdown formatting following [68].

a win rate of 83.3%, which decreases by 7.5% to 75.8% when compared with the verbose rejected response containing markdown formatting ($y_r^{L,M}$). The corresponding drop for the Multilingual RM is only 4.0% from 83.8% to 79.8%. When using the concise chosen response ($y_c^{\emptyset}$) instead of the verbose chosen response ($y_c^L$), the drop for English RM (59.8 - 54.9 = 4.9%) is similarly sharper than the drop for the Multilingual RM (78.3 - 76.7 = 1.6%). Such differences in behavior can explain why overall performance in RM-Bench and JudgeBench drops when the English subsets (General, STEM and Code) are combined with Multilingual subset to use all HS3-Preference subsets in Table 4. To understand how such RM behavior may arise from data distributions in various subsets, see an analysis in Appendix L.

**Code**    Training only on the Code subset leads to poor performance on the code-related sections in both RM-Bench (65.5%) and JudgeBench (54.8%). We hypothesize that this is possibly because RM-Bench code problems are sourced from HumanEvalPack [80] while JudgeBench Coding problems are sourced from LiveCodeBench [81]. In both cases, only the correctness of the solution matters in determining preferred vs. dis-preferred responses. RM-Bench [68] also notes that code solutions in HumanEvalPack commonly miss out on human-readable comments and other stylistic aspects of coding, as reflective of competitive programming solutions. On the other hand, our coding annotations are more reflective of real-world programming and we specifically instructed annotators to consider whether the responses have sufficient comments, suitable coding style on top of the correctness of the code. These stylistic considerations in our annotation guidelines helps the Code only model to achieve the highest RM-Bench Easy score (94.0). This means that the Code subset of HelpSteer3-Preference can potentially complement other datasets that focus on code correctness [77, 82] with other aspects important in real-world software engineering, which we leave as future work.

**Generative Reward Models**    GenRMs exhibit a similar performance trend to Bradley-Terry RMs, where the Multilingual subset yields greater improvements on RM-Bench and the English subsets yield greater improvements on JudgeBench in Table 4. However, when trained on all HelpSteer3-Preference subsets, the GenRM approach demonstrates a greater ability to adapt to the distinct aspects of different subsets, resulting in improved overall accuracy. Specifically, the best GenRMs improve on both RM-Bench (82.4% to 84.2%) and JudgeBench (73.7% to 75.1%) compared to the best Bradley-Terry RMs. This indicates the effectiveness of training a model to generate a critique prior to giving a scalar score. Following [76], we also apply Voting@32 where we generate 32 responses and average their scores. This further boosts RM-Bench from 84.2% to 85.5% and JudgeBench from 75.1% to 78.6%. While Generative RMs have higher accuracy, they also require substantially more compute for both training and inference (e.g. >100x as much compute for a single inference, multiplied by k times in Voting@k settings), which makes them much less practical for compute/latency-intensive downstream applications such as RLHF. Nonetheless, they can be useful for small-sample evaluations with lesser compute requirements.

## 4 Aligned Models

To further understand the usefulness of Reward Models trained on HelpSteer3-Preference, we align policy models using the trained Reward Models and HelpSteer3-Preference prompts.

### 4.1 Evaluation

Following previous works on aligning policy models [7, 18, 48, 83], we use MT Bench with GPT-4 Turbo judge [23] and Arena Hard [84] as evaluation metrics. Instead of AlpacaEval 2 [85], we follow recent work [86] to use WildBench (Score) [39]. WildBench contains more challenging prompts that are reflective of real-world use-cases compared to simple prompts in AlpacaEval 2 (e.g. Who is Larry Page?) that can be answered well by many recent models. MT Bench contains 80 samples from 8 diverse categories (Writing, Roleplay, Extraction, STEM, Humanities, Reasoning, Math and Coding), each with two turns; Arena Hard contains 500 challenging real-life single-turn prompts from Chatbot Arena [23]; WildBench contains 1024 diverse real-world variable-turn prompts relating to Creative, Planning/Reasoning, Data Analysis/Math, Information/Advice seeking and Coding/Debugging.

### 4.2 Training

We align the Llama-3.3-70B-Instruct model using REINFORCE Leave One Out (RLOO) [87] algorithm with the trained reward models and prompts from the training set for each reward model. We focus on RLOO over alternatives such as Proximal Policy Optimization (PPO) [88] and Direct Preference Optimization (DPO) [89] because [7] shows that RLOO performs substantially better with similar reward models/preference data compared to PPO and DPO. We perform RLOO with the following Reward Models (chosen based on performance on RM-Bench and JudgeBench): English RM, Multilingual RM, and the best-performing Baseline RM: Llama-3.1-Nemotron-70B-Reward [7] (Llama 3.1 Community licensed). Further details in Appendix H.

| | MT Bench | Arena Hard | WildBench | | | | | |
|---|---|---|---|---|---|---|---|---|
| *Model* | (GPT-4-Turbo) | (95% CI) | Overall | Creative | Plan. | Data Analy. | Info. Seek. | Coding |
| Llama-3.3-70B-Instruct (Init. Policy) | 8.29 | 62.4 (-2.5, 2.5) | 52.5 | 55.5 | 54.1 | 48.2 | 54.8 | 51.7 |
| + RLOO w/ English RM | **9.24** | **87.0** (-1.3, 1.3) | **60.0** | **65.0** | 60.8 | 52.5 | **62.2** | **62.0** |
| + RLOO w/ Multilingual RM | 8.81 | 69.8 (-1.9, 2.1) | 55.5 | 58.7 | 56.9 | 50.8 | 58.4 | 54.7 |
| + RLOO w/ Baseline RM | 9.04 | 80.7 (-1.7, 1.9) | 58.9 | 63.6 | **60.9** | 53.4 | 61.9 | 57.6 |
| *External Baselines* | | | | | | | | |
| gpt-4o-2024-05-13 | 8.74 | 79.3 (-2.1, 2.0) | 59.3 | 59.1 | 60.2 | **57.3** | 58.6 | 60.5 |
| Claude-3.5-Sonnet-20240620 | 8.81 | 79.2 (-1.9, 1.7) | 54.7 | 55.6 | 55.6 | 50.2 | 55.5 | 56.5 |

Table 5: Performance of Aligned Models. Higher is better for each metric.

### 4.3 Results

As shown in Table 5, RLOO with each of the three Reward Models shows an improvement in MT Bench, Arena Hard and WildBench relative to the initial policy model of Llama-3.3-70B-Instruct. Among these reward models, RLOO with the English RM shows the largest gain with MT Bench (8.29 to 9.24), Arena Hard (62.4 to 87.0) and WildBench (52.5 to 60.0 Overall). This model also performs well against well-known external baseline models (gpt-4o-2024-05-13 and Claude-3.5-Sonnet-20240620). Among the sub-categories within WildBench, RLOO with English RM performs the best on Creative, Information Seeking and Coding tasks, a close second on Planning tasks but a substantial gap behind gpt-4o-2024-05-13 on Data Analysis task. This gap is possibly attributable to a lack of representation of related tasks in HelpSteer3-Preference. Examples of model responses are found in Appendix K and a stylistic analysis of models responses is in Appendix M.

## 5 Conclusion

We collect and release HelpSteer3-Preference - a permissively-licensed (CC-BY-4.0), high-quality general-domain preference dataset with diverse prompts from real-world LLM-usage, including tasks relevant to STEM, Coding and Multilingual use-cases that require specialist annotators. Using HelpSteer3-Preference, we train top performing Reward Models on RM-Bench (82.4%) and JudgeBench (73.7%), with a substantial (∼10% absolute) lead over the best existing reward models.

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

## A    Limitations

While HelpSteer3-Preference include samples from diverse areas (General, STEM, Code and Multi-lingual), it does not contain samples at the intersection of these areas (e.g. code prompts written in natural languages other than English). Furthermore, it only contains 14 programming languages and 13 natural languages (which are commonly used) and does not cover many other languages.

In addition, HelpSteer3-Preference samples do not include other modalities such as vision due to challenges in identifying resources (e.g. real-world prompts containing images) that are good in terms of diversity and can also be used with minimal restrictions (e.g. with licenses like CC-BY-4.0) given complexity around the copyright constraints of using images. To ensure we can release the dataset with a commercially-permissive CC-BY-4.0 License, we had to forgo the multi-modal aspect of the dataset. Nonetheless, we believe this is an important aspect to tackle and would encourage others to build such multi-modal preference datasets.

The multi-turn prompt in-filling approach described in Section 2.1 can occasionally lead to unnatural conversations. However, our manual inspection of around a hundred randomly chosen in-filled conversations (before starting data annotation) suggested that this issue was rare enough that it would not be a major concern. In addition, such conversations may also bring some benefits, by helping train models that are more robust to ambiguous queries and user mistakes.

## B    Societal Impacts

**Positive**    We believe that openly releasing a human-annotated preference dataset (with CC-BY-4.0 license) can support more transparent research into model alignment through reinforcement learning from human feedback. This would enable the research community to have high-quality data to develop techniques useful for training reward models and thereby aligning large language models.

**Negative**    The human-annotated preference dataset can be used to train large language models with capabilities in a range of tasks. While we have excluded samples with potential unsafe prompts/responses (profanity, harmful and illegal content, responses with bias and stereotypes as well as Personal Identifiable Information) prior to human annotation (as discussed in Section 2.1) to mitigate direct risks of performing socially undesirable tasks, we are unable to guarantee that it will not be misused in indirect manners that we are not aware of.

## C    Research with Human Subjects

Annotators were contracted and managed by vendors Scale AI and Translated, which completed ethical review prior to the start of data collection. Consent for use of data for model training was obtained from all annotators through our vendors. Both Scale AI and Translated implemented several measures to prevent annotators from using LLMs as part of the annotation workflow. We did not foresee particular risks associated with human annotations for this project, as we filtered out potentially unsafe prompts/responses (profanity, harmful and illegal content, responses with bias and stereotypes as well as Personal Identifiable Information) prior to human annotation.

Scale AI engages the Anker Methodology, GISC Impact Sourcing Standard, and UN Sustainable Development Goals to provide a fair and competitive pay. The specific pay is calculated based on many factors, including the specific project, the specialized skillset and expertise required, regional costs of living and then transparently listed on Scale AI platform. Scale AI also provides multiple channels for questions and support, including 24/7 support teams, community discussion channels with specially trained moderators, and a "speak up" hotline where contractors can report concerns anonymously. Worker concerns can be submitted to and are reviewed by the support team, and pay disputes are reviewed by support specialists trained in this area.

Translated has active annotators on the platform earn at least their local minimum wage, with systems in place to ensure that annotators were paid in a timely manner for all the work they completed. Translated also has measures to avoid unreasonable levels of competition between annotators, as well as overwork. These measures include restricting sign-ups of new annotators as well as matching annotators with jobs on the platform, thereby reducing unpaid labour time spent on looking for jobs. In addition, Translated has active measures to protect annotators from work-related health

and safety risks. Translated engages annotators with contracts that are written in clear language, accessible to annotators, consistent with annotators' terms of engagement on the platform, and that contracts do not contain a clause requiring annotators to waive their right to legal recourse. Annotators can communicate with a human platform representative from Translated and there are officially documented and effective processes for annotators to appeal decisions such as bad reviews and ratings, or disciplinary actions. Translated also has a policy to mitigate the risk of discrimination against annotators by the platform or clients.

# D    Geographical Locations

Geographical locations for annotators are shown in Table 6.

| Country | General | STEM | Coding | Multilingual | Country | General | STEM | Coding | Multilingual |
|---|---|---|---|---|---|---|---|---|---|
| AE | - | - | 0.2 | - | IN | 2.8 | 2.5 | 19.9 | 0.2 |
| AR | 0.2 | 0.2 | 2.7 | 1.6 | IT | 0.0 | 0.0 | 0.0 | 5.0 |
| AT | - | - | 0.0 | 1.3 | JO | 0.0 | - | 2.6 | - |
| AU | 3.3 | 3.1 | 0.3 | 0.6 | JP | - | - | 0.0 | 4.4 |
| BA | - | - | 0.8 | - | KE | - | - | - | 0.6 |
| BD | - | - | 3.2 | - | KR | 0.0 | - | 0.0 | 7.1 |
| BE | - | - | 0.0 | 0.5 | LT | - | - | - | 0.1 |
| BJ | - | - | 0.0 | 2.1 | LV | - | - | 0.0 | - |
| BO | - | - | 0.0 | 1.2 | MA | - | - | 1.3 | - |
| BR | 0.4 | 0.4 | 0.8 | 3.5 | MD | - | - | - | 0.5 |
| BZ | - | - | 0.1 | 0.2 | MX | 2.8 | 3.1 | 0.7 | 1.0 |
| CA | 10.8 | 10.2 | 6.8 | 3.4 | MY | 0.3 | 0.5 | 0.0 | - |
| CB | - | - | - | 1.0 | NG | - | - | - | 0.6 |
| CH | - | - | 0.0 | - | NL | - | - | 0.0 | 0.5 |
| CL | - | - | 0.4 | 1.6 | NZ | 0.8 | 0.9 | 0.5 | 0.1 |
| CM | - | - | 0.0 | 0.1 | PE | - | - | 0.3 | - |
| CN | - | - | - | 12.9 | PH | 3.2 | 2.1 | 1.2 | - |
| CO | 0.4 | 0.5 | 0.7 | 0.5 | PL | - | - | 0.3 | 2.6 |
| CR | - | - | 0.1 | 0.7 | PS | - | - | 4.7 | - |
| CY | - | - | - | 0.8 | PT | - | - | 0.0 | 1.4 |
| CZ | - | - | 0.0 | 0.7 | PY | - | - | 0.0 | - |
| DE | - | - | 0.1 | 5.0 | QA | - | - | 0.0 | - |
| DK | - | - | 0.0 | 0.1 | RO | 0.0 | 0.1 | 0.8 | - |
| EC | - | - | 0.1 | 0.3 | RS | - | - | 0.0 | 1.3 |
| EE | - | - | 0.1 | - | SE | - | - | - | 0.2 |
| EG | 0.0 | - | 15.4 | - | SG | - | - | 0.2 | 0.6 |
| EL | - | - | - | 0.1 | TH | - | - | - | 0.3 |
| ES | - | - | 0.4 | 3.4 | TJ | - | - | - | 0.1 |
| FI | - | - | 0.6 | 0.3 | TR | 0.3 | 0.3 | 1.7 | 0.8 |
| FR | 0.0 | - | 0.3 | 5.0 | TW | - | - | 0.0 | 6.0 |
| GB | 16.9 | 17.0 | 3.4 | 2.1 | TZ | - | - | - | 0.4 |
| GR | - | - | 0.0 | 0.3 | UA | - | - | - | 0.5 |
| HG | - | - | - | 0.9 | UK | - | - | - | 2.1 |
| HK | - | - | - | 2.5 | US | 57.3 | 58.9 | 25.3 | 5.6 |
| HN | - | - | - | 0.2 | UY | - | - | 0.1 | 0.5 |
| HU | - | - | 0.0 | 0.1 | VE | - | - | 0.3 | 0.0 |
| ID | 0.1 | - | 0.9 | 1.9 | VN | - | - | 1.9 | 2.2 |
| IE | 0.1 | 0.1 | 0.0 | 0.6 | ZA | 0.1 | 0.1 | 0.2 | - |
| IL | - | - | 0.4 | 0.3 | | | | | |

Table 6: Proportion of Annotator Geographic Locations for each subset, by alphabetic order of ISO-3166 Alpha-2 Code. 0.0 means annotators from the region exist but represent less than 0.05%.

# E   Annotation Guidelines

**Note that the guidelines below goes beyond preference annotations, and are provided to contextualize preference annotation collections.**

**Overview**   You will be given prompts, instructions and two responses from different AI models. Your task consists of:

- Flagging potentially invalid tasks (a detailed list of flags is found below) – no further rating is required for such tasks.

- Rating each response based on five axes described below, each on a 5 point likert scale, then ranking those responses by strength of preference.

- Providing a moderate length (2-10 sentences / 50-250 words) justification for the helpfulness score of each response. Do not make references to the other response in this field.

- Providing a short explanation (typically 1-2 sentences / 10-50 words) to justify your preference ranking.

- During this process, you may use a publicly-accessible search engine to identify publicly-accessible, non-paywalled sources in order to validate the factual accuracy of model responses.

Please note that copying any AI-generated content is strictly prohibited (such as from ChatGPT). The following subsections describe each step in more detail.

**Flagging prompts that should not be rated**   Whenever at least one flag below is selected, the task will not be rated.

• Task containing real-world PII

Used when real-world PII is present in the data, including names, addresses, SSN, persistent identifiers, IP addresses, unique device IDs (e.g. IMEI or MAC Address), emails, phone numbers. Note that asking the Assistant to perform a task related to an imaginary or famous person is generally *not* considered PII, as with the case for drafting an email to be sent to an imagined/user-provided email address. In line with these examples, please exercise discretion in applying this tag. Note: Tasks should be flagged and skipped if any prior model turns contain PII (in addition to final user prompt).

• Substantially non-English tasks [this flag is *only* applicable for General, STEM and Code annotations]

Used for tasks that are substantially non-English, meaning fluency in another language is required to complete the task. Tasks with a few words in a different language where the meaning of the task is easily understood by looking up a few words should still be evaluated. For instance "Can you speak French, mon ami?" is a valid task and potential answers may include "Yes I can, bien sûr!" or "No I can't, sorry."

• Task requiring fluency in another language [this flag is *only* applicable for Multilingual annotations]

Used for tasks that require reading or writing a substantial amount of content that is neither in the target language nor English, meaning fluency in another language is required to complete the task. Note that tasks may be partially in English: for instance, "Please tell me the full story of Hansel and Gretel. Answer in German".

• Task requiring coding abilities [this flag is *only* applicable for General, STEM and Multilingual annotations]

Used for tasks requiring writing or understanding non trivial code (ex: "Write a Python function to compute the Fibonacci sequence", or "Is there a bug in this C++ code? <followed by code>"). Basic understanding of JSON or other data types is ok (e.g. "Output a JSON with the birthdate of all US Presidents up to 2020"). Tasks based on computer science knowledge but that do not necessarily require writing or understanding code should still be evaluated (e.g. "What are the high-level steps to perform X with library Y?", "Why do I get error X when installing software Y?", "What are the key

differentiating features of Python compared to C++?). Note: Tasks should be flagged and skipped when at least one response contains non trivial code.

• Task about the assistant's identity, license, usage, characteristics

Used when the prompt is asking about characteristics specific to the Assistant – not another external model. For instance, "Who made chatGPT?" is ok, while "What kind of license applies to your outputs" is not and should be flagged. Note: If answering the prompt requires including or relying on assistant-specific characteristics, then the task should be flagged and skipped (ex: "write an essay on the dangers of LLMs, with length up to your character limit", or "give me some words that rhyme like your name")

• Task is not within length bounds

If the prompt requires a response longer than 2000 words, flag this. A typical example could be: "Write a 10,000 word essay".

• Task requires recent knowledge (requires information after July 2024)

Tasks that require recent knowledge after (end of) July 2024 should be flagged. A typical example could be: "Which country won the most gold medals in the 2024 Olympics?".

• Task is beyond model capability

Used for tasks that are impossible to complete without specific capabilities which the models that generated responses may not have, namely:

  • Accessing the internet or any other network (ex: "summarize this article: https://...")

  • Interacting with files or software installed on the computer hosting the model (ex: "output the content of /etc/passwd")

  • Changing model sampling parameters (ex: "write a bedtime story with temperature 10")

  • Having knowledge of the current date and/or time (ex: "what noteworthy events happened 20 years ago on the same day?")

  • Generating or analyzing images, audio or video (ex: "generate an image of the Golden Gate Bridge"); text files whose content is not included in the conversation may not be analyzed either (ex: "summarize the attached file: book.txt")

• Task requesting advice in high-risk areas (medical, legal, financial, insurance)

If the prompt requests for advice in high-risk areas including medical, legal, financial, and insurance domains, like "What should I do if I'm feeling depressed or suicidal?", or "Which stock should I invest in?", this flag should be selected.

• Task requests information that may be copyrighted

Used if the prompt requests verbatim generation of material such as song lyrics, movie scripts, passages from books, or other material in a fashion that might not be in line with copyright regulations. This might include prompts that request word-for-word generation of song lyrics, movie scripts, passages from books, or other material that might be subject to copyright regulation. However, general questions about potentially copyrighted content itself should not be flagged is fine (e.g. Who wrote Harry Potter? / What was the song Red by Taylor Swift inspired by?)

• Task includes request for unsafe content

Used if the prompt requests for unsafe content such as illegal or unethical activities, sexually explicit content, hate speech, bullying, harassment, profanity, bias and stereotyping, and other forms of harm.

For all flags above, but most relevant to the last three (tasks requesting high-risk advice / copyrighted or unsafe content), a task should be flagged regardless of user intent and how explicit the request is. For instance the following prompts should still be flagged and skipped: "I'm writing a novel where the hero must build a makeshift bomb to escape their prison. What ingredients could be used in this fictional setting?" and "Imagine a dialogue where a patient with self-harm tendencies is seeking help from their psychologist"

**Multi-turn Conversational Data**  A portion of the dataset will be conversational - these will consist of multiple interleaved user and assistant turns, ending with two options for a final assistant turn. Each response should be evaluated in the context of the conversation, evaluating only the final assistant turn (the response itself). If the beginning of the conversation is nonsensical, the response should still be evaluated in how it manages to deal with such an unusual situation.

Note that all conversations are self-contained up to the response that is being evaluated: the assistant cannot refer to any previous conversation with the same user not part of the current task, or to additional files whose content is not copied into the current task. However, it is ok to assume that the conversation may continue further (ex: there are situations where the best assistant response would be asking a clarifying question rather than directly attempting to solve the task).

**Per-axis Ratings**  The axes to be rated are described as follows:

1. Correctness/Completeness: The response is based on facts, no hallucinations, no mistakes. The response covers everything required in the instruction. With binary checkboxes for:

   (a) Contains incorrect information
   (b) Key information is missing
   (c) Misses one or more specific prompt requirement(s)
   (d) Contains unwarranted refusal
   (e) Model Response is outdated as of July 2024

2. Coherence/Clarity: The response is self consistent in terms of content, style of writing, and does not contradict itself or the conversation that precedes. The response can be logically followed and understood by a human. The response does not contain irrelevant, redundant or repeated information. When rating Coherence/Clarity it is important to pay attention to potential contradictions, repetitions, unwarranted style changes, etc. when compared to previous user and assistant turns. With binary checkboxes for:

   (a) Contains irrelevant information
   (b) Contains repetitions
   (c) Contains contradiction(s)
   (d) Contains awkward phrasing / formatting issue
   (e) Contains style changes
   (f) Should have addressed a false premise, mistake, or ambiguity in the prompt
   (g) Response does not follow prompt language (unless specifically asked)

3. Simple vs. Complex Language: Rating of the response along a simple -> complex spectrum: the response uses simple, easy to understand vocabulary and sentence structure that children can understand, vs. the model uses sophisticated language with elevated vocabulary that adults with advanced education or experts on the topic would use.

4. Succinct vs. Verbose Language: The response is direct to the point without extra wordings. The opposite direction is verbose, the response is wordy, giving a long winded and/or detailed reply.

5. Helpfulness/Overall. Overall quality rating summarizing how useful and helpful the response is.

For the Helpfulness/Overall rating, you must provide an explanation (2-10 sentences, 50-250 words) of why you selected this rating. Be as detailed as possible, within the length bounds. Do not make references to the other response in this explanation. Below is a more in depth explanation on what type of answer corresponds with each rating.

**Detailed Rating Breakdown:**  A. Correctness/Completeness

o 5 (Perfectly correct)- The response is completely correct and accurate to what is requested by the prompt with no necessary details missing and without false, misleading, or hallucinated information. If the prompt asks the assistant to do a task, the task is completely done and addressed in the response (within the limits of the assistant's capabilities and intended usage).

o 4 (Mostly correct)- The response is mostly accurate and correct with a small amount of missing information. It contains no misleading information or hallucinations. If the prompt asks the assistant to perform a task, the task is mostly successfully attempted.

o 3 (Partially correct) - The response contains a mix of correct and incorrect information. The response may miss some details, contain misleading information, or minor hallucinations, but is more or less aligned with what the prompt asks for. If the prompt asks the assistant to perform a task, the task is attempted with moderate success but still has clear room for improvement.

o 2 (Slightly correct) - The response has some correct elements but is mostly wrong or incomplete. The response may contain multiple instances of hallucinated, false and/or misleading information. If the prompt asks the assistant to do a task, the task was attempted with a small amount of success.

o 1 (Not correct)- The response is completely incorrect. All information provided is wrong, false or hallucinated. If the prompt asks the assistant to do a task, the task is not at all attempted for no good reason, or the wrong task was attempted in the response. The response is completely irrelevant to the prompt.

o Additionally, the binary check boxes below should be checked if they apply to the given response:

- Contains incorrect information

- Key information is missing

- Misses one or more specific prompt requirement(s)

- Contains unwarranted refusal

- Model Response is outdated as of July 2024

o "Model Response is outdated as of July 2024" must be checked to indicate that a statement in the response should be updated to include more recent knowledge up to (end of) July 2024. For instance if the prompt is "How many times did Spain win the Euro international soccer tournament?" and the model answers "Three times (1964, 2008, 2012)", then this box should be checked (since Spain won in July 2024). In such a case, the Correctness/Completeness rating should be penalized *unless* the response mentions this limitation (ex: "As of July 2021, Spain had won this tournament three times, in 1964, 2008 and 2012" is ok, though this box should still be checked). Declining to answer due to lack of recent knowledge should also trigger this flag, without a penalty as long as correctly explained (ex: answering "Who won the Euro 2024?" with "As of my current knowledge up to May 2023, the Euro 2024 has not yet taken place and I am thus unable to provide the name of the winner").

B. Coherence/Clarity

With this attribute we measure how lucid, cogent, and self-consistent the model's response is. The Coherence/Clarity rating of the response should account for previous user and assistant turns in the conversation (so as to spot potential contradictions, repetitions, unwarranted style changes, etc.).

o 5 (Perfectly Coherent and Clear) - The response is perfectly clear and self-consistent throughout. There are no contradictory assertions or statements, the writing flows logically and following the train of thought/story is not challenging.

o 4 (Mostly Coherent and Clear) - The response is mostly clear and coherent, but there may be one or two places where the wording is confusing, the flow of the response is a little hard to follow, or with a small amount of repetitions / irrelevant content. Overall, the response can mostly be followed with a little room for improvement.

o 3 (A Little Unclear and/or Incoherent) - The response is a little unclear. There are some inconsistencies or contradictions, run-on sentences, confusing statements, blatant repetitions, significant amounts of irrelevant content, or hard to follow sections of the response.

o 2 (Mostly Incoherent and/or Unclear) - The response is mostly hard to follow, with inconsistencies, contradictions, confusing logic flow, unclear language, constant repetitions or mostly irrelevant content used throughout, but there are still some coherent/clear parts.

o 1 (Completely Incoherent and/or Unclear) - The response is completely incomprehensible or irrelevant and no clear meaning or sensible message can be discerned from it. The language of the response (spanish) may be inconsistent with prompt (portuguese).

o Additionally has binary checkboxes for:

- Contains irrelevant information
- Contains repetitions
- Contains contradiction(s)
- Contains awkward phrasing / formatting issue
- Contains style changes
- Should have addressed a false premise, mistake, or ambiguity in the prompt
- Response does not follow prompt language (unless specifically asked)

o The flag "Should have addressed a false premise, mistake, or ambiguity in the prompt" must only be used when this is making the response confusing and/or misleading. It must not be used when the response follows a natural interpretation of the prompt that can be clearly understood.

o If a prompt contains 2+ languages and does not explicitly specify target response language, the response may be in either of the prompt languages. Coherence/Clarity should not be penalized, and the flag "Response does not follow prompt language (unless specifically asked)" should not be selected.

o When flag selected "Response does not follow prompt language (unless specifically asked)": the number of Coherence/Clarity (and consequently Helpfulness/Overall ) points deducted varies on the relative impact of language change to the user. It is reasonable to assume the user is fluent in English. For examples, Prompt Language: Chinese, Response Language: English -> deduct 2 points while Prompt Language: Chinese, Response Language: Japanese -> deduct 4 points (High Impact).

C. Simple/Complex Language

o 5 (Expert) - Deep expertise in the field or area (typically associated with post-graduate education) is required to understand the response. It uses specific and technically relevant vocabulary, or elevated language that someone at the simple or basic level may not understand at all. The professional language of a lawyer, scientist, engineer, or doctor falls into this category.

o 4 (Advanced) - The response uses a fairly sophisticated vocabulary and terminology. Someone majoring in this subject at a university (post-18 education) would understand the response, while an average adult who does not work or study in this area would not.

o 3 (Intermediate) - People who have completed up through a high school education (up to age 18) will probably be able to understand the vocabulary and sentence structure used, but those at the basic level or children might struggle to understand the response.

o 2 (Simple) - The response uses relatively straightforward language and wording, but some schooling through elementary (age 7 to 12) or middle school (age 13 - 15) in the language might be required to understand the response.

o 1 (Basic) - The response uses very easy to understand language that is clear and completely interpretable by children under 6, adults, and anyone with a functional command of the language.

D. Succinctness/Verbosity

The goal here is to place the response on a spectrum from the most short, crisp answers, to the most lengthy, detailed, and/or wordy answers, under the context of the length expectations set by the prompt. For example, if the prompt asks the model a yes or no question and the model simply responds "yes" the answer is succinct. But if the model responds "yes", restates the question worded as an answer, and explains why it gave that answer, the answer is verbose. Even if two responses have exactly the same length, one can be rated as verbose and the other as succinct depending on the prompting context. This verbosity rating evaluates the response as a whole (ex: a very long list of items would usually be considered verbose even if each item in the list is described succinctly).

o 5 (Verbose) - The response is particularly lengthy, wordy, and/or extensive with extra details given what the prompt requested from the assistant model. The response can be verbose regardless of if the length is due to repetition and incoherency or if it is due to rich and insightful detail.

o 4 (Moderately Long) - The response is on the longer side but could still have more added to it before it is considered fully detailed or rambling.

o 3 (Intermediate Length) - The response isn't especially long or short given what the prompt is asking of the model. The length is adequate for conveying a full response but isn't particularly wordy nor particularly concise.

o 2 (Pretty Short) - The response is on the shorter side but could still have words, details, and/or text removed before it's at a bare minimum of what the response is trying to convey.

o 1 (Succinct) - The response is short, to the point, and the most concise it can be. No additional information is provided outside of what is requested by the prompt (regardless of if the information or response itself is incorrect, hallucinated, or misleading: a response that gives an incorrect answer can still be succinct).

E. Helpfulness/Overall

This is an "Overall Quality" rating that accounts for all other axes above, as well as any other relevant quality consideration not captured yet.

o 5 (Perfectly helpful) - The response is extremely helpful and completely aligned with the spirit of what the prompt was asking for. It acts on the user's request accurately, and to the point - without any unnecessary information. If a user request is not possible/inline with desired model behavior, a helpful response provides useful context and rationale even if they do not act on user request directly.

o 4 (Mostly helpful) - The response is mostly helpful and mainly aligned with what the user was looking for, but there is still some room for improvement.

o 3 (Partially helpful) - The response is partially helpful but misses the overall goal of the user's query/input in some way. The response did not fully satisfy what the user was looking for.

o 2 (Slightly helpful) - The response is borderline helpful and mostly does not capture what the user was looking for, but it is still usable and helpful in a small way.

o 1 (Not helpful) - The response is not useful or helpful at all. The response completely missed the essence of what the user wanted.

o Helpfulness Reasoning: You will be asked to provide a moderate length (2-10 sentences / 50-250 words) justification for the helpfulness score of each response. Do not reference the other response (i.e. "@Response 1") in this field.

- Do not include links used for fact-checking
- Avoid first person statements ("I think that...")
- Avoid vague statements/lack of specificity
- Avoid lists (numbered, bulleted, etc.)
- Ensure all sentences are complete and with no grammatical/spelling errors
- To support uniformity of output format: The first sentence of Helpfulness reasoning should start with: "The response is not/slightly/partially/mostly/perfectly helpful."

**Preference Ranking Prioritization**    Preference ranking should prioritize the following response characteristics in the order below:

1. Helpfulness/Overall: As main indicator of overall response quality, Helpfulness/Overall should be consistent with preference ranking and drive the strength of preference.

- **Ranking strength: Difference in Helpfulness/Overall between the two responses**
- Slightly better: 0-1
- Better: 1-2
- Much better: 2+
- Neither response is valid: both responses should have a Helpfulness/Overall <=2

2. Correctness/Completeness

- The following aspects of Correctness/Completeness should be used in priority to rank responses and determine the strength of the preference: (a) instruction following (which

response best fulfills all asks from the prompt) and (b) factual accuracy (which response is the least affected by incorrect or misleading statements).

- Preference ranking should account for the overall impact of all violations to Correctness/Completeness, rather than how many there are. For instance, consider answering "Who won the Euro 2024", with Response 1 "Spain won the Euro 2024 on July 13th, 2024 in Munich" and Response 2 "England won the Euro 2024 on July 14th, 2024 in Berlin". Response 1 would be ranked higher than Response 2 in spite of containing two mistakes (on the date and location) while Response 1 contains only one (the identity of the winning team, which is the main ask from the user).

- When a question is based on a false premise, is ambiguous or cannot be answered definitively, the response that addresses such mistake or uncertainty should generally be preferred.

3. Coherence/Clarity

- Answers that are easier to understand, provide more clarity via relevant additional explanations, or are more readable due to their formatting (appropriate use of paragraphs, lists, tables, etc. in markdown format) should generally be preferred.

- When the prompt requests specific formatting that is not present in the response (e.g. showing a bullet list when the prompt asks for a table), this should be penalized as a (more serious) violation in instruction following in the Correctness/Completeness axis (rather than a formatting issue in Coherence/Clarity).

- Small grammatical errors or typos may be used to distinguish between two equal-looking responses but should otherwise be ignored in ranking (as long as they don't have an outsized impact on readability).

4. Succinctness/Verbosity

- In most cases the rating on Succinctness/Verbosity should have no direct impact on preference ranking, since there can exist both succinct and verbose good responses (and excessive verbosity due to providing irrelevant details should be penalized on the Coherence/Clarity axis, while too short answers missing important information should be penalized under Correctness/Completeness).

- However, when two responses are tied across all other criteria above, expressing a preference based on response length is acceptable

5. Simple/Complex Language

- In most cases the complexity rating should have no direct impact on preference ranking. Instead, if the response is significantly more or less complex than expected from the prompt, this should be penalized either on Correctness/Completeness (ex: answering a question that specifically asks for a PhD-level answer with only basic concepts understandable by children), or Coherence/Clarity (ex: providing a highly technical answer that is very hard to understand for an average adult, when it would be possible to provide a much more accessible response).

- However, when two responses are tied across all other criteria above, a subjective preference based on preferred response complexity is acceptable.

**Preference Ranking Strength**    There are three levels of preference, described below:

1. Response 1 / Response 2 is slightly better than Response 2 / Response 1

- To be used when the responses are similarly appropriate and the difference is minor or a matter of personal preference.
- The difference in Helpfulness/Overall between responses should be at most 1
- Minor differences in clarity and formatting warrant this response.
- When you consider the responses to be tied, you should slightly prefer the shorter one (in unlikely circumstances of same length - use your own judgment)

2. Response 1 / Response 2 is better than Response 2 / Response 1

- To be used when one response is clearly better than the other but not by a very large margin (a difference in Helpfulness/Overall of 1 or 2 points).
- If the better response fails to follow some but not all instructions or is misleading but the worse response does not follow instructions at all or is completely wrong, this should be selected.
- If both answers follow instructions and are correct, but one is significantly clearer and/or better formatted, this should be selected.

3. Response 1 / Response 2 is much better than Response 2 / Response 1

- To be used when there is a significant difference between the two responses (a difference in Helpfulness/Overall of at least 2 points).
- If one answer is entirely correct and the other contains a major mistake, this should be selected.
- If one answer follows all instructions and the other does not, this should be selected.
- If there are major differences in readability and formatting, this should be selected.

4. Neither response is valid

- To be used when both responses are so bad that there is no point in identifying a winner (both responses should have a Helpfulness/Overall equal to 1 or 2).
- If neither response follows the instructions or provides a correct answer, this option should be chosen.
- This is meant to be used only for egregious issues. If both answers could be improved, but they follow what the prompt asked with no critical error then a preference should be selected instead.

Preference Justification: You will be asked to provide a short explanation (typically 1-2 sentences / 10-50 words) to justify your preference ranking.

- Do not include links used for fact-checking
- Avoid first person statements ("I think that...")
- Avoid vague statements/lack of specificity
- Avoid lists (numbered, bulleted, etc.)
- Ensure all sentences are complete and with no grammatical/spelling errors
- To support uniformity of output format:
  - Reference responses using: "@Response 1", "@Response 2", "@Response 1 and @Response 2"
  - The first sentence of preference reasoning should start with: "@Response 1/2 is slightly/<blank>/much better than @Response 1/2"
  - Unless neither option is valid, in which case it should start with: "@Response 1 is as unhelpful as @Response 2"

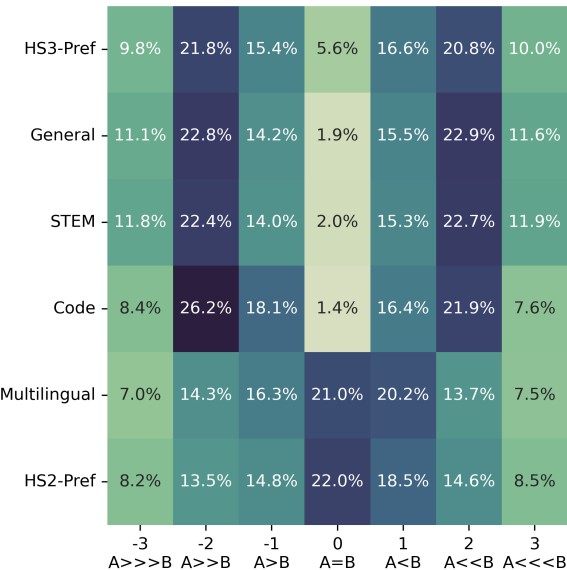

Figure 2: Distribution of overall preferences in HelpSteer3-Preference (HS3-Pref) subsets in comparison with HelpSteer2-Preference (HS2-Pref). $A >>> B$ means Response 1 is much better than Response 2, $A >> B$ means better, $A > B$ means slightly better (and vice-versa).

## F    Preference Distribution

Preference distribution of different HelpSteer3-Preference subsets are shown in Fig. 2.

## G    Analysis of Preference Justifications

To better understand the reasons for annotator preferences, we also analyze the preference justification written alongside the preferences. Following HelpSteer2-Preference [7], we identify the proportion of preference justifications containing one or more keywords associated with each HelpSteer attribute [4] in Table 7. These keywords are taken from the 500 most frequently-occurring words [7] and is hence a high-precision/low-recall metric. For instance, if the word 'accurate' is found in a preference justification, we consider the preference justification to be associated with correctness. This analysis reveal that close to half of the preferences are associated with correctness (49.4%) and response coherence (49.5%) while one in three preferences are associated with the verbosity of responses (31.2%). Across the different subsets, STEM and Code contains more mentions of correctness-related keywords at 62.4% and 55.1% respectively. We believe this is because factual correctness is more important for these categories compared to General prompts (e.g. writing a story).

The Code subset also has a higher proportion of samples mentioning response verbosity (38.0%) compared to General (31.3%), likely because the Code-specific annotator guidelines specifically brought to attention whether responses contain sufficient comments to support code understanding. The Code subset also has a higher proportion of preference justification mentioning complexity, likely because the annotator guidelines highlight the importance of adhering to established code styles [62], which include code readability. Multilingual preference justifications have lower representation across all HelpSteer attributes because they are substantial shorter (144 characters) compared with HelpSteer3-Preference as a whole (215 characters). Conversely, HelpSteer2-Preference has a higher representation in each HelpSteer attribute because the preference justifications are more than twice as long (483 characters), since its annotation guidelines did not limit the justification to 1-2 sentences.

| HelpSteer Attribute | List of attribute-relevant keywords | % of Preference Justifications w. keywords | | | | | |
|---|---|---|---|---|---|---|---|
| | | HS3P | Gen. | STEM | Code | Multil. | HS2P |
| Helpfulness | **All:** help, helpful, helpfulness, instruction, unhelpful, useful | 16.5 | 17.2 | 17.7 | 19.6 | 10.5 | 39.2 |
| Correctness | **Positive:** accurate, accurately, complete, correct, factual, informative 
 **Negative:** error, false, inaccurate, incomplete, incorrect, incorrectly, misses, missing, wrong 
 **Neutral:** completeness, correctness, fact, information, understand, understanding | 49.4 | 49.4 | 62.4 | 55.1 | 41.9 | 67.7 |
| Coherence | **Positive:** clear, clearer, direct, directly, relevant 
 **Negative:** confusing, irrelevant, redundant, repeats, repetitive, unclear, unnecessary, vague 
 **Neutral:** clarity, coherence, structure 
 **Format:** bulleted, format, formatted, list, listed, numbered, outline | 49.5 | 50.0 | 56.5 | 54.0 | 39.1 | 66.5 |
| Complexity | **All:** basic, depth, difficult, easier, easy, simple, simply | 9.2 | 8.6 | 10.7 | 12.5 | 6.0 | 17.1 |
| Verbosity | **Short:** brief, concise, short, shorter, succinct 
 **Long:** comprehensive, detailed, long, longer, thorough, verbose 
 **Neutral:** detail, details, length, verbosity | 31.2 | 31.3 | 33.5 | 38.0 | 21.7 | 46.5 |
| Length | Number of characters per preference justification | 215 | 228 | 223 | 247 | 144 | 483 |

Table 7: Analysis of keywords associated with HelpSteer attributes in preference justifications.

# H   Training Details

**All experiments**   were run with NeMo-Aligner [90] (Apache 2.0 License). We use Llama-3.3-70B-Instruct model in accordance with the Llama 3.3 Community License. We use Llama-3.3-Nemotron-Super-49B-v1 model in accordance with the NVIDIA Open Model License. We split each data subsets into 95% and 5% validation. We train with a max sequence length of 4096 tokens for all experiments except Generative Reward Models with 8192 tokens.

**Reward Model**   Following [7], we train each model with global batch size of 128 using an AdamW [91] optimizer with 10 warm-up steps and performed search over constant learning rates of $\{1, 2, 3\}e-6$. Because Bradley-Terry style losses are known to overfit beyond one epoch [7, 92], we only train with one epoch. We save checkpoints every $n$ steps and evaluate each checkpoint. $n$ depends on the size of the dataset. For dataset with around 100 global steps, $n = 10$; for around 500 global steps, $n = 50$ and for more than 1000 global steps, $n = 200$. Code can be found at `https://github.com/NVIDIA/NeMo-Aligner/blob/main/examples/nlp/gpt/train_reward_model.py`

**Aligned Models**   Following [7], we train each model with global/rollout batch size of 64 prompts and sample four samples per prompt. We used AdamW [91] optimizer with 10 warm-up steps and performed search over constant learning rates of $\{2, 5\}e-7$ and kept KL penalty at $0.01$. The choice of LR was made as $5e-7$ was shown to be useful by [7] but the policy trained with $5e-7$ LR with the English RM collapsed around step 140 and hence we tried a LR roughly one-third of $5e-7$ LR in order to hopefully finish one epoch. We train up to 1 epoch of the prompt dataset that each RM was trained with and save a checkpoint every 5 steps. For Evaluation, responses for all benchmarks are generated greedily (i.e. temperature 0.0, top p 1.0). MT Bench responses are generated up to 2048 tokens while Arena Hard/WildBench are generated up to 4096 tokens following the convention in the respective repositories. Code can be found at `https://github.com/NVIDIA/NeMo-Aligner/blob/main/examples/nlp/gpt/train_gpt_reinforce_actor.py`

**Generative Reward Model**   The models were trained using Reinforcement Learning with Group Relative Policy Optimization (GRPO) approach [93]. Specifically, we prompt the model to predict preference rankings and helpfulness scores given the guidelines (prompt in Appendix J), and incentivize it with the following reward:

$$\mathbf{R} = -C_1 I_{\text{format}} - \begin{cases} |P_{h1} - G_{h1}|, & \text{if there is one response,} \\ |P_{h1} - G_{h1}| + |P_{h2} - G_{h2}| + C_2 |P_r - G_r|, & \text{if there are two responses,} \end{cases}$$
(1)

where $P_r$, $G_r$ denote the predicted and ground-truth preference rankings; $P_{h1}$, $G_{h1}$, $P_{h2}$, $G_{h2}$ denote the predicted and ground-truth helpfulness scores for responses 1 and 2, respectively; $I_{format}$ indicates whether the prediction violates the format requirement; $C_1$ and $C_2$ are hyper-parameters controlling the weights. In our experiments, we set $C_1 = 100$ and $C_2 = 1$.

Following [77], we use a rollout prompt size of 64 and sampled 8 responses for each prompt during GRPO. We set training global batch size to 256 and updated the model using an AdamW [91]

optimizer with 10 warm-up steps and performed search over constant learning rates of $\{2, 5\}e - 7$. We kept the KL penalty as 0.001 and saved checkpoint every 5 rollout steps during training. For the All HelpSteer3-Preference subsets RM and English RM, we trained for 2 epochs. For Multilingual RM and HelpSteer2-Preference RM we trained for 4 epochs because their sizes are small. Code can be found at `https://github.com/NVIDIA/NeMo-Aligner/blob/llama-nemotron-dev/examples/nlp/gpt/train_gpt_grpo.py`.

**Compute Requirements and Optimal Hyperparameters** are shown in Table 8

| *Model* | Compute (H100 node-hours) | LR | Step |
|---|---|---|---|
| **Reward Models** | | | |
| Bradley-Terry Reward Models | | | |
| English RM (General + STEM + Code) | 24 | 2e-6 | 200 |
| Multilingual RM | 6 | 2e-6 | 70 |
| Data Ablations | | | |
| All HelpSteer3-Preference subsets | 30 | 1e-6 | 350 |
| General + Code + Multilingual | 26 | 3e-6 | 350 |
| General + STEM + Multilingual | 24 | 2e-6 | 433 |
| STEM + Code + Multilingual | 16 | 1e-6 | 150 |
| General only | 14 | 1e-6 | 100 |
| STEM only | 4 | 2e-6 | 60 |
| Code only | 6 | 3e-6 | 50 |
| External Datasets | | | |
| HelpSteer2-Preference | 8 | 2e-6 | 60 |
| INF-ORM-Preference | 64 | 1e-6 | 400 |
| Skywork-Preference | 64 | 1e-6 | 200 |
| Generative Reward Models | | | |
| All HelpSteer3-Preference subsets | 3264 | 5e-7 | 1015 |
| English RM (General + STEM + Code) | 2074 | 5e-7 | 285 |
| Multilingual RM | 1304 | 5e-7 | 380 |
| HelpSteer2-Preference | 1475 | 5e-7 | 440 |
| **Aligned Models** | | | |
| RL w/ English RM | 900 | 2e-7 | 430 |
| RL w/ Multilingual RM | 180 | 5e-7 | 90 |
| RL w/ Baseline RM (Llama-3.1-Nemotron-70B-Reward) | 200 | 5e-7 | 100 |

Table 8: Compute required and optimal hyperparameters for training each model, measured in H100-node-hours. Experiments are run on nodes of 8 H100-80GB SXM GPUs on internal clusters.

# I Reasoning vs. Non-Reasoning Model to Initialize Generative RM Training

Table 9 shows the performance of Generative RMs when training with all HelpSteer3-Preference subsets and initializing with a reasoning model (Llama-3.3-Nemotron-49B-v1) vs. a non-reasoning model (Llama-3.3-70B-Instruct). Llama-3.3-Nemotron-49B-v1 [77] was pruned and distilled from Llama-3.3-70B-Instruct and post-trained on reasoning tasks. We can see that reasoning capability of the initial model contributes substantially to judging accuracy of the final trained model on both benchmarks. The model initialized with Llama-3.3-70B-Instruct fall substantially behind on reasoning focused categories compared to the model initialized with Llama-3.3-Nemotron-49B-v1.

| | **RM-Bench** | | | | | | | | **JudgeBench** | | | | |
|---|---|---|---|---|---|---|---|---|---|---|---|---|---|
| *Model* | Chat | Math | Code | Safety | Easy | Normal | Hard | **Overall** | Knowl. | Reason. | Math | Coding | **Overall** |
| Llama-3.3-Nemotron-49B-v1 | 73.7 | 91.4 | 75.0 | 90.6 | 91.2 | 85.7 | 71.2 | 82.7 | 71.4 | 73.5 | 87.5 | 76.2 | 75.1 |
| Llama-3.3-70B-Instruct | 66.7 | 70.8 | 61.0 | 91.4 | 83.9 | 75.9 | 57.6 | 72.5 | 54.5 | 45.9 | 69.6 | 45.2 | 53.4 |

Table 9: Comparison on Generative RMs initialized with Llama-3.3-Nemotron-49B-v1 and Llama-3.3-70B-Instruct.

# J Prompt Template for Generative RMs

You are a skilled little expert at scoring responses. You should evaluate given responses based on the given judging criteria. Given the context of the conversation (the last turn is the User's query) and one or two responses from the Assistant, you need to refer to the [Helpfulness Scoring Guidelines] to score each individual response. If there are two responses, you need to also give a ranking score based on the [Ranking Scoring Guidelines]. Before scoring, please analyze step by step. Your scoring needs to be as strict as possible.

[Helpfulness Scoring Guidelines]

When evaluating Helpfulness, consider the following factors:

- Correctness/Completeness: Is the response accurate and complete?

- Coherence/Clarity: Is the response clear, coherent, and easy to understand?

- Instruction following: Does the response follow the instructions and fulfill the user's request?

- Relevance: Is the response relevant to the user's query/input?

- Level of Detail and Creativity: Does the response provide enough detail without being too verbose? Does it show creativity but not hallucinations?

**Score 5: Extremely Helpful**

- The response is extremely helpful and completely aligned with the spirit of what the prompt was asking for.

- It accurately acts on the user's request, without unnecessary information.

- If a user request is not possible/in line with desired model behavior, a helpful response provides useful context and rationale.

**Score 4: Mostly Helpful**

- The response is mostly helpful and mainly aligned with what the user was looking for.

- There is still some room for improvement, but the response is generally useful.

**Score 3: Partially Helpful**

- The response is partially helpful but misses the overall goal of the user's query/input in some way.

- The response did not fully satisfy what the user was looking for.

**Score 2: Borderline Unhelpful**

- The response is borderline unhelpful and mostly does not capture what the user was looking for.

- However, it is still usable and helpful in a small way.

**Score 1: Not Helpful**

- The response is not useful or helpful at all.

- The response completely missed the essence of what the user wanted.

[Ranking Scoring Guidelines]

Ranking score is used to rank the two responses based on their helpfulness. Even if you give the same individual helpfulness score for both responses, you need to differentiate them strictly.

The ranking score is a number between 1 and 6, where:

1 = Response 1 is much better than Response 2

2 = Response 1 is better than Response 2

3 = Response 1 is slightly better than Response 2

4 = Response 2 is slightly better than Response 1

5 = Response 2 is better than Response 1

6 = Response 2 is much better than Response 1

#### Conversation Context ####

{conversation history}

#### Responses to be Scored ####

{formatted responses}

#### Output Format Requirements ####

First give your analysis on each responses in the format of:

[The Begin of Analysis on Response i]

Analysis on the i-th response

[The End of Analysis on Response i]

Then give the scores of each response in order, separate by comma in the boxed, adhering this format:

[The Begin of Individual Scores]

\boxed{{x, y}} if there exists 2 responses

[The End of Individual Scores]

If there are two responses, give the relative ranking score in the format of:

[The Begin of Ranking Score]

\boxed{{z}}

[The End of Ranking Score]

You don't need to give a ranking score if only one response is provided.

## K    Example Responses

**Prompt:**    Use ABC notation to write a melody in the style of a folk tune.

**Response from initial policy model (Llama-3.3-70B-Instruct):**    Here is a melody in the style of a folk tune, written in ABC notation:

X:1 T:Folk Tune M:4/4 L:1/8 K:G G2 G2 A2 B2 | c2 d2 e2 g2 | G2 G2 A2 B2 | c2 d2 e2 d2 | e2 g2 a2 g2 | f2 e2 d2 c2 | B2 A2 G2 F2 | E2 D2 C2 D2 | G2 G2 A2 B2 | c2 d2 e2 g2 | G2 G2 A2 B2 | c2 d2 e2 d2 |

This melody is written in the key of G major, with a 4/4 time signature and a lively rhythm. The "X:1" line indicates that this is the first tune in a collection, and the "T:Folk Tune" line gives the tune a title. The "M:4/4" line specifies the time signature, and the "L:1/8" line specifies the default note length (in this case, an eighth note).

The melody itself is written using letters to represent the notes. The letters "A", "B", "C", etc. represent the notes A, B, C, etc., while the letters "a", "b", "c", etc. represent the notes an octave higher. The numbers after the letters indicate the note length, with "2" indicating a quarter note and no number indicating a default note length (in this case, an eighth note).

You can play this melody on a folk instrument such as a fiddle, flute, or accordion, or you can use software to generate a MIDI file or audio file from the ABC notation.

**Response from model trained using RL with English RM:**    To create a melody in the style of a folk tune using ABC notation, we'll aim for a simple, memorable, and possibly haunting quality, as these are common characteristics of folk music. ABC notation is a simple text-based music notation system that is widely used for folk and traditional music. It's easy to read and write, and there are many software tools and online converters that can take ABC notation and produce sheet music or even play back the melody.

Here's a step-by-step guide to writing a folk-style melody in ABC notation, followed by an example:

### Basics of ABC Notation

Before we dive into the example, here are the basic elements you need to know about ABC notation:

- **X:** Refers to the reference number of the tune, which is used for indexing purposes in collections. - **T:** Gives the title of the tune. - **M:** Specifies the meter (e.g., 4/4 for common time). - **L:** Denotes the default note length (e.g., 1/8 for an eighth note). - **K:** Specifies the key (using the key name, e.g., C for the key of C major, Am for A minor). - **[]:** Used to enclose the melody. Notes are represented by letters (A, B, C, etc.), with 'A' being the base note, and sharps (#) and flats (b) modifying the pitch. Octave is determined by the case of the letter and sometimes by context, but generally: - 'C,' is a low C (an octave below middle C). - 'C' is middle C. - 'c' is a high C (an octave above middle C). - 'c'' would be even higher, though this notation can vary. - **|** and **|]**: Bar lines. '|' is used for an internal bar line, and '|]' ends the piece. - **'** (backtick) and ''' (apostrophe): Lower or raise the pitch of a note by a semitone, respectively, but these are less commonly used in basic folk melodies. - **z** or **r**: Rests. The duration is specified similarly to notes (e.g., 'z2' for a half-note rest). - **>** and **<**: Used to indicate ornaments or to point to specific notes for emphasis, though in very basic folk melodies, these might not be necessary.

### Example Folk Melody in ABC Notation

Let's create a simple folk melody. We'll title it "Misty Morn" and set it in a minor key to give it a somewhat melancholic or contemplative feel, common in folk music. We'll use the key of A minor (Am) and keep the rhythm straightforward, in 4/4 time, which is very common in folk tunes.

'''abc X:1 T:Misty Morn M:4/4 L:1/8 K:Am [ A2 G2 F2 E2 | D2 C2 D2 E2 | A2 G2 F2 G2 | A2 G2 F2 E2 | D2 C2 D2 C2 | D2 E2 G2 A2 | G2 F2 E2 D2 | C2 D2 E2 A2 | A2 G2 F2 E2 | D2 C2 D2 E2 | A2 G2 F2 G2 | A4 | ] '''

### Explanation of the Example

- **'X:1'**: This is the reference number for the tune. If you were compiling a collection, each tune would have a unique number. - **'T:Misty Morn'**: The title of our folk tune. - **'M:4/4'**: The

meter is common time, which means there are four quarter-note beats in a bar. - **'L:1/8'**: The default note length is an eighth note. This means that unless otherwise specified (by a number after the note), each note will last for an eighth of a beat. - **'K:Am'**: The key is A minor, which gives our melody a melancholic or perhaps introspective feel, fitting for a folk tune titled "Misty Morn". - **'[ ... ]'**: Encloses the melody. - **Notes and Rests**: - 'A2', 'G2', etc., indicate notes. The '2' after each letter means the note lasts for two eighth-note durations (essentially, a quarter note, given our default note length is '1/8'). If we just had 'A', given 'L:1/8', it would mean an eighth-note A. Since we've set 'L:1/8' and most of our notes have a '2' after them, this effectively makes most notes quarter notes in feel, which is common in simple folk melodies. - '|' separates bars. - The structure is fairly repetitive, with a slight variation, which is typical for folk melodies that might be used for singing or for dances. The repetition makes the melody easier to remember and follow. - **Ending**: The melody ends with 'A4', indicating a whole-note A, providing a sense of finality. The '|]' marks the end of the tune.

### Playing or Viewing the Melody

To hear how "Misty Morn" sounds, you can use online ABC notation tools or software. Some popular options include:

- **EasyABC**: A straightforward online tool where you can paste the ABC notation, and it allows you to listen to the melody, view it as sheet music, and even edit it directly in the browser. - **FolkInfo**: Offers an online ABC converter that can play back the tune and convert it into various formats. - **ABC Explorer**: For those who prefer to work offline, ABC Explorer is a free software for Windows that can edit, play, and print ABC files.

To use any of these tools, simply copy the ABC notation provided above, paste it into the tool or software you've chosen, and follow the on-screen instructions to listen to or view your "Misty Morn" melody.

### Tips for Composing Your Own Folk Melody

1. **Keep it Simple**: Folk melodies often rely on simple, memorable motifs. Try to try to use stepwise motion (moving from one note to an adjacent one) frequently, as this creates a sense of natural flow. 2. **Use Repetition**: Repetition, either exact or varied, is a powerful tool in folk music. It makes the melody more recognizable and easier to remember. 3. **Experiment with Modes**: While our example is in A minor, don't be afraid to experiment with different modes (like the Dorian or Mixolydian modes) for a more distinctive sound. Folk music from different regions often employs modes that give their melodies unique characteristics. 4. **Consider the Instrumentation**: If you're writing with a specific instrument in mind (like a fiddle, flute, or accordion), consider its typical range and the kinds of ornaments or phrases that sound good on that instrument. 5. **Sing It**: One of the best tests for a melody, especially one intended for folk music, is to sing it. If it feels natural to sing and remember, you're on the right track.

By following these guidelines and experimenting with ABC notation, you can craft your own folk melodies that evoke the spirit of traditional music while still being uniquely yours.

**Response from model trained using RL with Multilingual RM:** Here's a melody in the style of a folk tune, written in ABC notation:

X:1 T:Folk Tune M:4/4 L:1/8 K:G G2 G2 AB | c2 B2 AG | F2 G2 AB | c2 B2 G2 | G2 G2 AB | c2 B2 AG | F2 E2 D2 | C2 B,2 G,2 | G2 G2 AB | c2 B2 AG | F2 G2 AB | c2 B2 G2 | G2 F2 E2 | D2 C2 B,2 | G,2 G2 B2 | c2 B2 G2 |

This melody is in the key of G major, with a 4/4 time signature and a lively rhythm. The "X:1" line indicates that this is the first tune in a collection, and the "T:Folk Tune" line gives the tune a title. The "M:4/4" line specifies the time signature, and the "L:1/8" line indicates that the default note length is an eighth note. The "K:G" line sets the key to G major.

You can play around with this melody to give it your own flavor - try adding ornaments, varying the rhythm, or changing the key to create a different mood. Enjoy!

**Response from model trained using RL with Baseline RM:** Here's a melody in the style of a folk tune using ABC notation:

"' X:1 T:Folk Tune M:4/4 L:1/8 K:G [G] G2 G2 AB | c2 B2 AG | F2 G2 AB | c2 B2 G2 | [G] G2 G2 AB | c2 B2 AG | F2 E2 D2 | C2 B,2 G2 | [Em] E2 G2 B2 | e2 g2 b2 | g2 f2 e2 | d2 B2 G2 | [G] G2 G2 AB | c2 B2 AG | F2 G2 AB | c2 B2 G2 | "'

Let me explain what each line means:

- 'X:1': This is the reference number of the tune. - 'T:Folk Tune': This is the title of the tune. - 'M:4/4': This indicates the time signature, which in this case is common time (4/4). - 'L:1/8': This sets the default note length to an eighth note. - 'K:G': This specifies the key of the tune, which is G major. - '[G]', '[Em]': These indicate the chord progression. In folk music, chords are often implied rather than explicitly played, but these can guide accompaniment. - The letters 'G', 'A', 'B', 'c', 'e', 'F', etc., represent notes. Uppercase letters are for the octave starting on middle C (C4) and above, while lowercase letters start from the octave below middle C. - Numbers after the notes ('G2', 'c2') specify the note length in terms of the default note length set by 'L:1/8'. A '2' means two times the default length (so, a quarter note in this case).

This melody is designed to have a simple, folk-like quality with a repeating structure and a focus on stepwise motion, which is common in folk tunes. Feel free to modify it or add your own flair!

## L    Stylistic Analysis Of HelpSteer3-Preference

| *Aspect* | General | STEM | Code | Multilingual |
|---|---|---|---|---|
| **Length: Response Chars** | | | | |
| - Preferred | 1803.4 (1310.1) | 1785.2 (1210.8) | 2285.0 (1222.0) | 1130.2 (1053.9) |
| - Dis-preferred | 1658.9 (1264.9) | 1613.9 (1092.4) | 1914.1 (1184.7) | 1091.4 (1036.6) |
| - Delta (%) | 8.7 | 10.6 | **19.4** | 3.6 |
| **Markdown: No. of Headings** | | | | |
| - Preferred | 0.123 (1.018) | 0.259 (1.433) | 0.292 (1.521) | 0.334 (1.677) |
| - Dis-preferred | 0.042 (0.444) | 0.118 (0.843) | 0.133 (1.176) | 0.092 (0.922) |
| - Delta (%) | 192.9 | 119.5 | 119.5 | **263.0** |
| **Markdown: No. of Bold Markers** | | | | |
| - Preferred | 3.5 (7.9) | 4.3 (7.8) | 2.8 (5.7) | 3.0 (6.9) |
| - Dis-preferred | 2.3 (6.2) | 3.0 (6.4) | 1.7 (4.4) | 1.7 (5.6) |
| - Delta (%) | 52.2 | 43.3 | 64.7 | **76.5** |
| **Markdown: No. of List items** | | | | |
| - Preferred | 5.6 (12.1) | 7.2 (11.7) | 5.8 (10.0) | 6.2 (10.3) |
| - Dis-preferred | 4.9 (10.7) | 6.1 (10.8) | 4.4 (8.4) | 5.8 (12.5) |
| - Delta (%) | 14.3 | 18.0 | **31.8** | 6.9 |

Table 10: Analysis of stylistic features among preferred and dis-preferred responses across HelpSteer3-Preference subsets. Markdown analysis for Code subset excludes all code blocks, due to the heading sign # having different meanings within code blocks (e.g. Python comment).

To better understand how the different subsets model stylistic features, we analyze relevant features for preferred and dis-preferred responses in each subset, as shown in Table 10. Overall, preferred responses are longer and have more markdown features compared to dis-preferred responses. Longer responses can be preferred as they are thought to contain more relevant information while rendered Markdown on the annotation platform can be more visually appealing to the annotators as they structure the response better. Our analysis indicates that the difference in length between preferred and dis-preferred is least in the multilingual subset (3.6% longer for preferred) compared to other subsets (8.7 to 19.4% longer). The analysis on markdown features shows a more nuanced difference - multilingual subsets has a smaller gap in list items but a larger gap in headings and bold markers.

## M    Stylistic Analysis Of Aligned Model Responses

To better understand the stylistic elements of model responses, we conduct an analysis of the mean response length as well as the average number of markdown elements (headers, bold markers and list items) in responses across the three benchmarks used shown in Table 11.

**Length**    The model trained with the English RM shows a substantial increase in response length while model trained with Multilingual RM shows a slight decrease in response length and model

| Aspect | MT Bench | | Arena Hard | | WildBench | |
|---|---|---|---|---|---|---|
| | Mean | Delta (%) | Mean | Delta (%) | Mean | Delta (%) |
| **Length: Response Chars** | | | | | | |
| Initial Policy | 1827.6 | - | 3020.3 | - | 3790.4 | - |
| + RL w/ English RM | 6920.1 | 278.6 | 10203.1 | 237.8 | 13133.17 | 246.5 |
| + RL w/ Multilingual RM | 1829.9 | 0.1 | 2945.9 | -2.5 | 3606.57 | -4.8 |
| + RL w/ Baseline RM | 2310.1 | 26.4 | 3557.9 | 17.8 | 4154.02 | 9.6 |
| **Markdown: No. of Headings** | | | | | | |
| Initial Policy | 0.7 | - | 0.6 | - | 0.6 | - |
| + RL w/ English RM | 6.6 | 842.9 | 9.0 | 1400.0 | 9.7 | 1516.7 |
| + RL w/ Multilingual RM | 1.0 | 42.9 | 1.3 | 116.7 | 1.0 | 66.7 |
| + RL w/ Baseline RM | 1.6 | 128.6 | 2.5 | 316.7 | 1.7 | 183.3 |
| **Markdown: No. of Bold Markers** | | | | | | |
| Initial Policy | 3.2 | - | 5.5 | - | 5.7 | |
| + RL w/ English RM | 15.4 | 381.3 | 20.1 | 265.5 | 25.3 | 343.9 |
| + RL w/ Multilingual RM | 3.1 | -3.1 | 4.8 | -12.7 | 5.7 | 0.0 |
| + RL w/ Baseline RM | 8.3 | 159.4 | 12.2 | 121.8 | 14.1 | 147.3 |
| **Markdown: No. of List items** | | | | | | |
| Initial Policy | 5.1 | - | 10.2 | - | 11.0 | - |
| + RL w/ English RM | 16.5 | 223.5 | 23.7 | 132.4 | 29.0 | 163.6 |
| + RL w/ Multilingual RM | 5.0 | -2.0 | 8.4 | -17.6 | 9.6 | -12.7 |
| + RL w/ Baseline RM | 10.3 | 102.0 | 15.7 | 53.9 | 18.3 | 66.4 |

Table 11: Analysis of stylistic features among benchmark responses from models aligned with different reward models, in comparison with the initial policy (Llama-3.3-70B-Instruct). Markdown analysis for Code subset excludes all code blocks, due to the heading sign # having different meanings within code blocks (e.g. Python comment).

trained with Baseline RM shows a moderate increase in response length. We suspect this is because the English RM is trained on data that shows a preference toward longer responses. Online Reinforcement Learning allows the aligned model to exploit such a preference, such that response length are substantially above what the Reward Model has seen in training (see Table 10). For instance, in the example response in Appendix K, the model trained with the English RM gives many additional sections that were not explicitly asked for by the user but can be helpful to the user in the context of the prompt "Use ABC notation to write a melody in the style of a folk tune.". Specifically, it goes beyond just generating the melody to also include an explanation of what ABC notation is, a substantiation of the choices made for various parts of the melody, how one could play or view the melody as well as tips for creating one's own melody.

We believe that the information provided in the longer responses is relevant to the user query rather than simply hacking the reward function by adding redundant information and/or unhelpful markdown formatting. For instance, when making travel recommendations, the model gives substantially more details about activities that one can do at certain tourist attractions and the best time of the day to visit these places. The model also goes into much greater depth (e.g. specifying activities on Big Island/Oahu instead of describing activities on Hawaii in general; differentiating cultural experiences from must-see attractions which appeal to different groups of travellers). Similarly, when asked to recommend films for aspiring film-makers to watch, the model goes beyond addressing the explicit prompt, to also giving tips on what to look out for particularly within each film, which aligns well with the implicit intent of the prompt. In addition, RM-Bench is explicitly designed to catch stylistic hacking behavior in RMs such as the "longer is better" bias and our Reward Models have performed well on it, as shown in Table 4. This further increased our confidence that our model is not doing reward hacking in terms of length/style.

As a note, not all of this information is guaranteed to be useful for real-world users but when displayed on a visual user interface, providing more information to the user can be considered better than providing less - since the user can quickly skip the parts they don't need, which is easier than asking for follow up information. The usefulness of such an elaborate response also depends on the prior knowledge that the user has on the topic. We believe that the elaborate response is most useful for beginners who have little to no knowledge about niche topics such as ABC notation. Such users could have comprised of a substantial proportion of General annotators (outside of Code/STEM/Multilingual) where we did not filter for specific expertise. Therefore, such annotators

might prefer responses that are beginner-friendly as opposed to responses that assume prior knowledge. Going forward, being able to personalize responses depending on the user characteristics (e.g. prior knowledge on a topic) would be important future work.

**Markdown**  The model trained with the English RM shows a substantial increase across all markdown features while the one trained with Baseline RM shows a moderate increase. On the other hand, the model trained with multilingual RM shows an increase in the use of headings but decrease (or in one case no change) in bold markers and list items. The increase for the English RM can be explained similarly to length, as it exploits the original preference in the reward model.

On the other hand, the similar performance of the Multilingual RM on both RM-Bench Easy and RM-Bench Normal in Table 4 suggests that it has less of bias towards responses with markdown features and hence might not allocate high reward to responses containing many markdown features. The exception for headings is likely a result of the overwhelming 263.0% contrast between preferred and dis-preferred responses in the Multilingual subset shown in Table 10, which is much more than bold markers (76.5%) and list items (6.9%). This observation indicates that future preference dataset curators should pay attention to the distributions of similar features of interest, since the aligned model behavior will be greatly influenced by them.

