# OpenReview forum: "HelpSteer3-Preference: Open Human-Annotated Preference Data across Diverse Tasks and Languages"
_NeurIPS.cc/2025/Datasets_and_Benchmarks_Track — NeurIPS 2025 Datasets and Benchmarks Track poster_

### Official Review · Reviewer_YcN3 · 2025-06-15

**Ethics Flags:** Human rights (including surveillance)
**Rating:** 5
**Confidence:** 4

**Summary:**

This paper introduces HelpSteer3-Preference, a high-quality human-annotated preference dataset designed to enhance the training of general-domain, instruction-following language models using Reinforcement Learning from Human Feedback (RLHF). The dataset comprises over 40,000 samples across diverse tasks, including STEM, coding, and multilingual scenarios, addressing the need for high-quality, diverse, and commercially-friendly preference data.

**Additional Feedback:**

- Q1: Why the low mean preference suggests the low position bias?

- Q2: Why couldn't all HelpSteer3-Preference subsets achieve the best performance in Table 4?

**Dataset Code Accessibility:**

Partly

**Dataset Code Comments:**

Missing code for training reward, generative and aligned models.

**Ethical Considerations:**

No, there are no or only very minor ethics concerns

**Final Justification:**

Considering the rebuttal and discussions, I would like to maintain the score and recommend acceptance.

**Limitations Weaknesses:**

The division of the second and third generation of general-domain preference datasets is not clear.

**Strengths Contributions:**

- Diverse and High-Quality Dataset: The dataset covers a wide range of real-world applications and is constructed with high-quality annotations from specialist annotators, ensuring strong inter-rater reliability.

- Improved Reward and Aligned Models: Using HelpSteer3-Preference, the authors trained reward models that achieved top performance on RM-Bench (82.4%) and JudgeBench (73.7%), representing a significant improvement over previous models. Besides, the dataset seems to be beneficial for downstream alignment tasks.

- Open-Source Availability: The dataset is released under a permissive CC-BY-4.0 license, allowing for broad commercial use and fostering further research and development in the field. The dataset has made a certain influence in the community.

---

> ### Author Rebuttal · Authors · 2025-07-30
>
> Thank you for emphasizing the influence that our open-source, diverse and high-quality dataset has made on the community!
>
>
> > The division of the second and third generation of general-domain preference datasets is not clear.
>
> The demarcation of 2nd and 3rd generation of general-domain preference datasets is primarily based on release date. The second generation datasets (UltraFeedback, HelpSteer, Nectar) were released in Oct-Nov 2023 while the third generation datasets (Skywork, HelpSteer2-Preference, INF-ORM) were released in Oct-Dec 2024.
>
> > Ethics Flags: Human rights (including surveillance)
>
> We understand that the Human rights section was flagged due to the nature of human annotations. As demonstrated in Appendix C, and in adherence to NeurIPS code of ethics, we compensated annotators fairly in accordance with their local pay standards. Each of our data vendors has also implemented processes to ensure that annotators are able to carry out annotations in supportive settings.
>
> > Missing code for training reward, generative and aligned models.
>
> The code for training each of these three types of models were linked to publicly accessible code in Appendix H.
>
> > Q1: Why the low mean preference suggests the low position bias?
>
> Mean preferences are calculated by taking the average of the overall preference of each sample, which can take one value among -3 (A>>>B) , -2, -1, 0, 1, 2,  and 3 (B>>>A).  A very low mean preference (close to -3) would suggest strong preference for the first response shown to annotators while a very high mean preference (close to 3) would suggest preference for the second response shown to annotators. Conversely, a mean preference close to 0 suggests that in aggregate, annotators do not have a preference for responses in either position. Our mean preference is -0.003 shown in Table 2, indicating that position bias is very low.
>
> > Q2: Why couldn't all HelpSteer3-Preference subsets achieve the best performance in Table 4?
>
> We find that the English RM subsets (General, STEM and Code) have substantially different distributions of preference compared with the Multilingual RM subset. As we discuss in lines 275 -301, the English subsets contain a strong preference for responses which are more detailed and nicely-formatted with markdown while the Multilingual subset has a weaker preference for such responses. This is supported by the differential change in performance on RM-Bench Hard when such response styles are accounted for (see Fig 1 / Line 286). We also analyzed the ground-truth distribution of such stylistic elements (length/markdown features) in Table 10 and found substantial differences among the different subsets.
>
>
> By simply combining all HelpSteer3-Preference subsets in training Bradley-Terry Reward Model, the model can receive contradictory information during training about what features characterize a good response, and hence not be aligned on a common “north star” that it optimizes towards. Granted, this situation  is not ideal and characterizes the very challenge behind putting together a preference dataset across diverse domains with different distributions. Instead of shying away from this challenge, we hope to inform the community about our experience curating such a preference dataset so that others can build upon our work to potentially find ways to overcome such limitations.

---

> > ### Comment · Reviewer_YcN3 · 2025-08-04
> >
> > I appreciate the authors' detailed reply. They have addressed my concerns. I will keep my score.

---

### Official Review · Reviewer_Q5n2 · 2025-06-30

**Rating:** 4
**Confidence:** 3

**Summary:**

The authors release HelpSteer3-Preference dataset, which contains 40K+ human-annotated preference pairs and covers multiple tasks including General, STEM, Code and Multilingual. It is roughly 4 times larger than HelpSteer2 and the construction of this dataset involves 5400+ specialist annotators from 77 regions. The authors demonstrated high inter-rater reliability and also trained reward models using this dataset to improve the SOTA on RM-Bench and JudgeBench.

**Additional Feedback:**

See Limitation section above.

**Dataset Code Accessibility:**

Yes

**Dataset Code Comments:**

The dataset is available on Huggingface.

**Ethical Comments:**

The prompts data are using existing sources. No clear ethical concerns are found.

**Ethical Considerations:**

No, there are no or only very minor ethics concerns

**Final Justification:**

I am raising my score to 4 (Borderline accept). The authors clarified several concerns in the rebuttal, especially regarding RewardBench, position bias, and the Code subset’s performance. While some novelty aspects remain incremental, the scale and diversity of HelpSteer3-Preference offer value to the community.

**Limitations Weaknesses:**

1.	Although the author argued the limitations of RewardBench, I still think it would be valuable to present the results on RewardBench, which is a popular evaluation benchmark for reward models.
2.	The author mentioned the low position bias but did not provide any details about it.
3.	The Code subset gives a poor performance on the benchmarks, possibly showing the limited quality of this subset.
4.	The  HelpSteer2 dataset considers multi-attribute scoring and studies the weighting of attributes. Do the same results apply to HelpSteer3 (e.g. optimal weighting of attributes)? The authors might consider adding a discussion about the multi-attribute scoring in the main body, which I think is important.
5.   HelpSteer 3 offers substantial incremental improvements over HelpSteer 2, but its novelty is somewhat limited in terms of core methodology.

**Strengths Contributions:**

The dataset has a large scale and diversity (it contains 40K+ pairs, annotated from 6400+ human annotators from various countries and regions). The authors demonstrate significant improvement of their trained model using HelpSteer3 on the benchmarks. They also demonstrated detailed analysis and clear presentation of the data distribution and model performance results.

---

> ### Author Rebuttal · Authors · 2025-07-30
>
> Thank you for highlighting the large-scale and diverse nature of the HelpSteer3 dataset, as annotated by thousands of human annotators across 70+ regions!
>
> Most of the highlighted concerns are misunderstandings of our work rather than limitations or weaknesses of the paper, which we will elaborate upon individually below.
>
> 1. Given the limitations of RewardBench discussed in Line 214, we believe that presenting results on RewardBench would do more harm than good for readers of the paper, as readers might be misled into thinking that models are better or worse than their RewardBench performance suggests. As researchers in reward modelling (or any other areas of machine learning), it is our responsibility to identify and utilize evaluation benchmarks that best proxy for an unbiased estimation of models’ performance. Using a biased [1,2] benchmark only because its (past) popularity can collectively slow down progress in our field. Nonetheless, if the reviewer has specific rationales for disagreements with the discussion on RewardBench’s limitations, we are happy to clarify further.
>
> 2. We agree that the paragraph on low position bias did not make a clear link with Table 2, which substantiates our claim. In particular, the mean preference field is calculated by taking the average of the overall preference of each sample, which can take one value among -3 (A>>>B) , -2, -1, 0, 1, 2,  and 3 (B>>>A). A very low mean preference (close to -3) would suggest strong preference for the first response shown to annotators while a very high mean preference (close to 3) would suggest preference for the second response shown to annotators. Conversely, a mean preference close to 0 suggests that in aggregate, annotators do not have a preference for responses in either position. Our mean preference is -0.003, indicating that position bias is very low.
>
> 3. The Code subset has 8857 samples, which is more than both the Multilingual subset (8063 samples) and the STEM subset (4918 samples), indicating that the performance of the Code subset should not be attributed to its size. In addition, the Code only subset performs poorly on RM-Bench (73.5) but relatively well on JudgeBench (72.0 - which is second highest among BT models). Therefore, we believe that the Code subset has uneven performance with unique strengths and weaknesses, discussed in Line 302.
>
> 4. We emphasize that this work (HelpSteer3-Preference) is a follow-up to HelpSteer2-Preference (ICLR 2025). Both are preference-only datasets that only contain information on which of the two responses to a common prompt is preferred and by how much. Therefore, it does not contain multi-attribute scores as contained in HelpSteer2 (NeurIPS 2024), which can be used to train models with adjustable weights.
>
> 5. We agree that the core methodology of collecting data in this work is in many ways inspired by HelpSteer2-Preference (ICLR 2025). The main improvements in this work comes from the increase in the diversity of tasks and languages covered from the specialist domains including Code, STEM and Multilingual. However, we believe that characterizing this work as incremental can be unfair because such scaling human annotation to new domains required the research team to innovate on many fronts (e.g. adding new domain-specific guidelines and conducting extensive quality assurance for tasks across 14 programming and 13 natural languages). Such innovations allowed us to collect diverse and multilingual preference data, which had never been publicly available and can empower research in ways that were not possible previously.
>
> We hope our response has addressed your concerns and it will be great if you can reconsider your assessment of the paper.
>
> [1] reWordBench: Benchmarking and Improving the Robustness of Reward Models with Transformed Inputs (Meta, 2025)
>
> [2] HelpSteer2-Preference: Complementing Ratings with Preferences (ICLR, 2025)

---

> > ### Comment · Reviewer_Q5n2 · 2025-08-05
> >
> > Thanks for the response and clarification, which addresses some of my concerns. The rebuttal clarified my concerns about RewardBench, position bias, and the Code subset’s performance. The authors provide concrete evidence and rationale for their design choices. While the methodology builds on HelpSteer2, the scale, diversity, and domain-specific innovations of HelpSteer3 add value to the community. Therefore I will raise my score.

---

### Official Review · Reviewer_tcCv · 2025-06-30

**Rating:** 5
**Confidence:** 4

**Summary:**

This paper introduces HelpSteer3-Preference, a novel dataset for RLHF training and evaluation of LLMs.
Specifically, it contains over 40000 human-annotated preference pairs, covering diverse tasks (STEM, coding, and multilingual scenarios, in addition to general-purpose tasks). The paper's results demonstrate the dataset's effectiveness in training high-performing RMs and subsequent policies via RLH.

**Dataset Code Accessibility:**

Yes

**Dataset Code Comments:**

Dataset is open-sourced.

**Ethical Considerations:**

No, there are no or only very minor ethics concerns

**Final Justification:**

Accept as the rebutal succesfully answered all reviewers' concerns

**Limitations Weaknesses:**

- The main weakness is that the paper would be even better with additional ablations on the dataset, for example:
  * scaling laws showing RM performances as the RLHF dataset grows
  * the impact of using human annotators (versus AI annotators or humans without the degree selection for example) on the same prompt.
  * better quantiative analysis of the diversity of the prompts.

- Minors
  * usually, paragraph's name should end with a ".".
  * contexts are restricted to a maximum of 2000 words might not be enough to capture long term dependencies. And why is it 2638 in Table 2?
  * why the standard deviations are that large in Table 2?
  * is the Cohen's coefficient computed before or after filtering?

**Strengths Contributions:**

- Be given the importance of reliable/diverse preference datasets, open-sourcing such a dataset is very relevant for the community.
  * The dataset is large, with a focus on diversity, including diverse domains, multilinguality and multi-turn conversations?
  * The datasets seems to be of high quality thanks to the use of human annotators.
- The trained RMs are of very good quality, reaching top performance on established benchmarks like RM-Bench (82.4%) and JudgeBench (73.7%).
- Interesting ablations:
  * of the different biases of the datasets, notably formatting bias, and the length bias of the English RM.
  * The paper also incudes lot of details about human annoationa that can be useful for the future.
  * I also appreciated the comparison between Discrimnatice and Generative based RMs.

---

> ### Author Rebuttal · Authors · 2025-07-30
>
> Thank you for recognizing the importance of our reliable and diverse dataset to the community as well as the strength of our Reward Models trained on this dataset.
>
> > **Ablations to further strengthen the paper**
>
> 1. Our experiments do not suggest a linear relationship between RLHF dataset size and RM performance. Here, we present the model performance trained on the full HelpSteer3-Preference dataset at different steps (i.e. every 50 steps, which is around 10% of one epoch). JudgeBench and RM-Bench both increase with more data until step 350 (around 62% of the full dataset) before plateauing.
>
> **Steps**|**JudgeBench**|**RM-Bench**
> :-----:|:-----:|:-----:
> 50|65.7|68.3
> 100|65.7|72.2
> 150|68.0|75.1
> 200|68.0|74.4
> 250|67.1|74.9
> 300|67.1|78.4
> **350**|**68.9**|**78.5**
> 400|68.3|74.9
> 450|68.0|76.3
> 500|68.0|77.3
> 550|67.7|76.5
> 561|68.0|77.9
>
> 2. Thank you for this suggestion. While it’s not practical for us to re-annotate data with a new group of human annotators without degree selection, we tried using AI annotators to re-annotate the data. Specifically, we prompted Llama 3.3 70B Instruct with the prompt template from Appendix J. We chose this model rather than closed-source models because of potential licensing/terms of use considerations. Our first observation was that the AI annotator suffers from substantial position bias seen from the distribution (mean preference of -0.7937, meaning strong position bias for Response 1) while human-annotated data has essentially no position bias (mean preference of -0.003). Mean preferences are calculated by taking the average of the overall preference of each sample, which can take one value among -3 (A>>>B) , -2, -1, 0, 1, 2,  and 3 (B>>>A). Detailed distributions of preference (in proportions, with each row summing to 100%)  are below, showing that the AI annotator has a strong bias for predicting Response 1 is much better than Response 2.
>
> Annotator | -3 (A>>>B) | -2 (A>>B) | -1 (A>B) | 0 | 1(A<B) | 2 (A<<B) | 3 (A <<<B)
> :------------:|:---------------:|:------------:|:---------------:|:-------:|:---------------:|:------------:|:----------:|
> Human     | 9.8 | 21.8 | 15.4 | 5.6 | 16.6 | 20.8| 10.0|
> AI  | 36.0 | 11.0 | 14.2 | 6.3 | 5.3 | 22.0 | 5.1 |
>
> Using this AI Annotator data, we replicated the training recipe for both the English RM and the Multilingual RM. The results in the table below, suggests that training with AI-Annotated data leads to much lower performance on both benchmarks (>5% absolute drop on JudgeBench and >10% drop on RM-Bench). This indicates that recruiting specialist Human-annotators as opposed to applying AI annotators substantially improved data quality.
>
> **Name**|**JudgeBench**|**RM-Bench**
> :---------------:|:-----:|:-----:
> English RM | 73.7 | 79.9
> English RM AI-Annotated | 67.1 | 68.7
> Multilingual RM | 69.4 | 82.4
> Multilingual RM AI-Annotated | 64.3 | 70.0
>
> 3. In addition to the diversity of prompts covering 14 Programming and 13 Natural Languages covered within the Code and Multilingual subsets from Table 3, we conduct an additional quantitative analysis to demonstrate the diversity of prompts. Specifically, we analyzed the prompts in the STEM subset on the university-level subject knowledge that is required to answer them. We follow the approach used to identify languages required to answer prompts, by prompting Nemotron-4-340B Instruct. The table below shows the 10 most common subjects, with a long tail of 370 subjects in total.
>
> **Subject**|**Proportion**
> :---------------:|:-----:
> ComputerScience | 12.5
> Physics | 5.4
> Chemistry | 4.6
> Mathematics | 4.2
> Engineering | 3.3
> Astronomy | 3.0
> Medicine | 2.5
> Statistics | 2.3
> Biology | 1.9
> Economics | 1.8
>
> > **Minors**
> 1. Thank you, we will add fullstops after paragraph titles.
> 2. 2638 refers to the average number of characters in the context in Table 2, which is roughly 500 words, if we assume each word has around 5 characters. This limit of 2000 words was set with the following consideration:
>
> - By allowing 2000 words to the context and another 2000 words to the response, there is a total of 4000 words, which approximately is the length of a NeurIPS paper.
>
> - We want annotators to fully read the response and when necessary, fact-check relevant claims within the article. By giving responses longer than our requirement to human annotators risks human annotators skimming through. There are potential work-arounds but these make the annotation exercise much more complicated.
>
> - Some of the models that we used for response generation (e.g. Nemotron 4 340B Instruct)  were only trained with context length of up to 4k tokens, which is around 4000 words and we did not want to risk generating low quality responses beyond the stated context length.
>
> 3. This question seems underspecified about which standard deviations are large. There are three distinct types of standard deviations in Table 2:
>
> - Context Turns: Some conversations only have a single-turn prompt while others might have 10 context turns. Our prompt-sampling approach tries to have a balanced spread of prompts with different lengths and hence results in a somewhat large variance.
>
> - Context/Response Chars: Similarly, there are user questions that are short (e.g. What is Power Gum) and those that are long (e.g. complex coding questions with substantial context requirements), which require answers of different lengths.
>
> - Mean Preference: This is calculated using the micro-average of preferences across all samples, each of which is -3 (A>>>B), -2, -1, 0, 1, 2 or 3(A<<<B). As a result, the variance (1.95) of such a distribution with a range of 6 is reasonable.
>
> 4. Cohen’s Kappa Coefficients in Table 2 are after-filtering. Filtering outlier annotations is key to our recipe for obtaining quality annotations as Cohen’s kappa are only around 0.5 prior to filtering.

---

### Official Review · Reviewer_J7rn · 2025-07-05

**Rating:** 5
**Confidence:** 4

**Summary:**

The paper introduces HelpSteer3-Preference, a permissively licensed human preference benchmark that succeeds HelpSteer2 and improves upon avaliable datasets by increasing task diversity, incorporating multilingual samples, and  relying on vetted specialist human annotation instead of general population or LLM-based preferences. Tasks include STEM, Coding, and 13 different non-English languages. They select from a wide variety of models for responses including Nemotron 4 (340B Instruct), Gemma (2B), Gemma 2 (2B, 9B, and 27B), Mistral (7B-Instruct-v0.3), Mistral-Nemo (12B), Codestral (22B), Mixtral (8x7B Instruct and 8x22B Instruct), Mistral Large 2, Phi 3 (Mini, Small, and Medium), IBM Granite (8B and 34B), and Snowflake Arctic.

The authors use this dataset to train both conventional (Bradley-Terry) and generative reward models (RMs), achieving new state-of-the-art performance on RM-Bench and JudgeBench with ~10% absolute gains over previous models. To demonstrate end-to-end utility, they align a Llama-3.3-70B model using the RLOO algorithm and their best RM, showing performance gains on established benchmarks like MT-Bench and Arena Hard that make it competitive with leading proprietary models.

**Dataset Code Accessibility:**

Yes

**Dataset Code Comments:**

Dataset and code release is of a very high quality, with great documentation and examples.

**Ethical Comments:**

The paper clearly addresses and mitigates concerns.

**Ethical Considerations:**

No, there are no or only very minor ethics concerns

**Final Justification:**

The authors have expanded upon my relatively minor issues regarding reward hacking and multimodality, I maintain my position for the acceptance.

**Limitations Weaknesses:**

- The use of multimodality has drastically increased since the original HH-RLHF and Open Assistant release, with a large portion of chats containing at least one image. This omission may limit the dataset's long-term relevance. While text-only data is still immensely valuable, the lack of multimodal examples is a clear limitation in scope. It would be interesting to know the challenges of adopting the existing pipeline to multimodal settings.

- The length of the responses has increased significantly, as seen in Table 11, while the proposed explanation of "contain more relevant information while rendered Markdown on the annotation platform can be more visually appealing to the annotators" is a reasonable, it could also be a stylistic reward hacking. The paper should investigate if this length adds real value or is just the model exploiting a "longer is better" bias

- Unfair comparison between RM types. The Generative RMs get a huge compute boost from Voting@k, but the Bradley-Terry models don't get any similar ensemble method. This makes the comparison of architectures feel unbalanced. Could a similar approach also boost the Bradley-Terry models?

- The exclusion of A=B samples is motivated, but considering their large amount, being 21% multilingual tasks it would be interesting to see if excluding them introduces any biases to the evaluation.

**Strengths Contributions:**

The work presents a significant contribution to the field, some notable strengths are:
- High quality of the raters and multiple ratings per sample, with approach of throwing outlayer voter and going by the 3 voters consesus and the resulting inter-rater reliability scores, as shown in table 2 give strong confidence for the quality of the resulting dataset.

- State of the art performence on the most challenging benchmarks - RM-Bench and JudgeBench, with gains of 10% over the exisitng models.

- Real world applicability on a downstream task as seen with the Llama-3.3-70B RLOO alignment results on MT Bench, Arena Hard and WildBench.
- Great breakdown, with judge models trained on specific subtasks and evaluated for different tasks as seen in the table 4. Excellent and easy-to-follow analysis of the observed trends.

-  Variety and usage of strong competitive models while avoiding restrictive license output such as GPT-family.

- Excellent Presentation - The paper is exceptionally well-written and structured. With a strong main text and supplementary material. The supporting materials and the dataset itself are well organized and ready for use.

---

> ### Author Rebuttal · Authors · 2025-07-30
>
> Thank you for appreciating our work and highlighting the quality of our dataset in terms of various aspects!
>
> Below, we address each of the questions the reviewer posed under Limitations Weaknesses.
>
> 1. We agree that multimodal data would be useful for the community. The main challenge lies in identifying resources (e.g. real-world prompts containing images) that are good in terms of diversity and can also be used with minimal restrictions (e.g. with licenses like CC-BY-4.0) given complexity around the copyright constraints of using images. Therefore, in order to ensure that we are able to release this resource with a commercially-permissive CC-BY-4.0 License, we had to forgo the multi-modal aspect of the dataset.
>
> 2. We had a similar concern after noticing this trend initially and only formulated our proposed explanation after manually looking through some examples within our evaluation sets. We believe that the information provided in the longer responses is relevant to the user query rather than simply hacking the reward function by adding redundant information and/or unhelpful markdown formatting. For instance, when making travel recommendations, the model gives substantially more details about activities that one can do at certain tourist attractions and the best time of the day to visit these places. The model also goes into much greater depth (e.g. specifying activities on Big Island/Oahu instead of describing activities on Hawaii in general; differentiating cultural experiences from must-see attractions which appeal to different groups of travellers). Similarly, when asked to recommend films for aspiring film-makers to watch, the model goes beyond addressing the explicit prompt, to also giving tips on what to look out for particularly within each film, which aligns well with the implicit intent of the prompt. In addition, RM-Bench is explicitly designed to catch stylistic hacking behavior in RMs such as the “longer is better” bias and our Reward Models have performed well on it, as shown in Table 4. This further increased our confidence that our model is not doing stylistic hacking, which we will take care to make clearer in our paper.
>
> 3. We did not apply Voting @ k to Bradley-Terry models because at inference time, Bradley-Terry models generate a scalar score that is invariant/fixed given a prompt and response, meaning that if one inference gives a score of 3.2, then multiple generations will all give the same score 3.2. Therefore, it is not possible to create useful “ensembles” with a single model. On the other hand, Generative RMs require temperature-sampling of thinking traces, which are non-deterministic, allowing it to benefit from Voting @ k (following DeepSeek GRM’s approach [1]).
>
> 4. We clarify that we only excluded A=B samples for training and not evaluation (which does not contain such tied samples to begin with). The reason is that Bradley-Terry style training does not natively support ties, meaning that we have to exclude all A=B samples.
>
> [1] Inference-Time Scaling for Generalist Reward Modeling (DeepSeek, 2025)

---

> > ### Comment · Reviewer_J7rn · 2025-08-07
> >
> > Regarding multimodality, perhaps expanding using references to other materials would be an option.
> > Thank you for looking into the possibility of reward hacking. Please expand on it in the final manuscript if accepted.
> > The rebuttal addressed my points. I will maintain the score.

---

### Note · Authors · 2025-08-13

We are grateful for the reviewers recognizing the value of key aspects of our work, which include:

1. Diverse, High-quality, Reliable and Large-Scale Human-Annotated Data (J7rn, tcCv, Q5n2, YcN3)
2. SOTA Performance of Reward Models (J7rn, tcCv, Q5n2, YcN3)
3. Detailed and Insightful Analysis and Ablations  (J7rn, tcCv, Q5n2)
4. Open-Source Dataset that excludes use of proprietary models (e.g. GPT-*) enabling commercial use with minimal restrictions (J7rn, YcN3)
5. Clear Presentation and Structuring of Work  (J7rn, Q5n2)

We appreciate the reviewers' insightful questions and constructive feedback, and believe that their initial concerns have been addressed in our rebuttals.

Regarding Reviewer J7rn's follow up comment re. Multimodal data and possibility of reward hacking, we agree this is a relevant point to discuss and will expand upon it in our final manuscript if accepted.

---

### Decision · Program_Chairs · 2025-09-18

**Decision:**

Accept (poster)

**Comment:**

This work introduces HelpSteer3-Preference, a large-scale, human-annotated preference dataset designed to advance the quality and diversity of openly available data for RLHF training, spanning over 40,000 samples across STEM, coding, and multilingual tasks. Reviewers highlighted the dataset’s scale, diversity, and reliability, the strong performance of resulting reward models, and the value of releasing an open-source, permissively licensed resource. In the rebuttal, the authors provided detailed clarifications on RewardBench usage, position bias, and Code subset performance, which addressed key reviewer concerns. While some novelty aspects are incremental, the overall contribution is timely, impactful, and of clear utility to the community. Considering the improved reviewer assessments and the strength of the contribution, the AC recommends acceptance.